# Contemporary HIV-1 consensus Env with AI-assisted redesigned hypervariable loops promote antibody binding

Hongjun Bai [1,2], Eric Lewitus [1,2], Yifan Li [1,2], Paul V. Thomas [2,3], Michelle Zemil [1,2], Mélanie Merbah[1,2], Caroline E. Peterson[2,3], Thujitha Thuraisamy[1,2], Phyllis A. Rees[2,3], Agnes Hajduczki[2,3], Vincent Dussupt [1,2], Bonnie Slike [1,2], Letzibeth Mendez-Rivera[1,2], Annika Schmid[1,2], Erin Kavusak[1,2], Mekhala Rao[1,2], Gabriel Smith[1,2], Jessica Frey [1,2], Alicea Sims [1,2], Lindsay Wieczorek[1,2], Victoria Polonis[1], Shelly J. Krebs [1], Julie A. Ake[1], Sandhya Vasan [1,2], Diane L. Bolton [1,2], M. Gordon Joyce[2,3], Samantha Townsley[1,2] & Morgane Rolland [1,2] ✉

An effective HIV-1 vaccine must elicit broadly neutralizing antibodies (bnAbs) against highly diverse Envelope glycoproteins (Env). Since Env with the longest hypervariable (HV) loops is more resistant to the cognate bnAbs than Env with shorter HV loops, we redesigned hypervariable loops for updated Env consensus sequences of subtypes B and C and CRF01_AE. Using modeling with AlphaFold2, we reduced the length of V1, V2, and V5 HV loops while maintaining the integrity of the Env structure and glycan shield, and modified the V4 HV loop. Spacers are designed to limit strain-specific targeting. All updated Env are infectious as pseudoviruses. Preliminary structural characterization suggests that the modified HV loops have a limited impact on Env's conformation. Binding assays show improved binding to modified subtype B and CRF01_AE Env but not to subtype C Env. Neutralization assays show increases in sensitivity to bnAbs, although not always consistently across clades. Strikingly, the HV loop modification renders the resistant CRF01_AE Env sensitive to 10-1074 despite the absence of a glycan at N332.

HIV-1 remains a global health priority. Vaccines have historically been proven to be the most powerful tool to protect against viral pathogens. Despite many setbacks in HIV-1 vaccine efficacy trials, the search for a safe and effective HIV-1 vaccine should continue. Several unique HIV-1 features have hampered the development of an HIV-1 vaccine. The protein that allows HIV-1 entry, Env, is metastable, heavily glycosylated, and highly sequence-diverse. To counteract HIV-1 diversity, broadly neutralizing antibodies (bnAbs) must be elicited by a vaccine to cross-react with a variety of Env[1,2]. The vaccine candidates tested in the nine

HIV-1 vaccine efficacy trials conducted to date did not yield bnAbs. Structural biology studies have revealed the atomic details of the epitopes recognized by numerous HIV-1 bnAbs (e.g. as reviewed[3–5]), providing a roadmap for designing antigens that could integrate these bnAb epitopes. Known HIV-1 bnAbs cover the Env surface[6] and are grouped in categories based on the location of their epitopes, including Env variable loop 2 (V2)-apex antibodies, glycan supersite antibodies, and CD4 binding site (CD4bs) antibodies[3–5]. Those bnAb epitopes are under functional constraints. For example, the V2-apex

[1]U.S. Military HIV Research Program, Walter Reed Army Institute of Research, Silver Spring, MD 20910, USA. [2]Henry M. Jackson Foundation for the Advancement of Military Medicine, Bethesda, MD 20817, USA. [3]Emerging Infectious Disease Branch, Walter Reed Army Institute of Research, Silver Spring, MD 20910, USA. ✉e-mail: mrolland@hivresearch.org

epitope shields the V3 loop and stabilizes the prefusion Env trimer[7,8] and the CD4bs epitope is where the host receptor binds to[9].

HIV-1 Env presents five variable loops and four of them (V1, V2, V4, and V5) contain a hypervariable segment, called the hypervariable (HV) loop. These HV loops can be adjacent to epitope sites recognized by bnAbs. The V1 HV loop (V1HV) is near the glycan supersite epitope (target of 10-1074[10], PGT135[11,12], PGT128[11,13] and 2G12[14]), the V2 HV loop (V2HV) is near the V2-apex epitope (target of PGT145[15–17], VRC26.25[18], PG9[19] and PGDM1400[15]), and the V5 HV loop (V5HV) is near the CD4bs epitope (target of VRC01[20] and 3BNC117[21]). These HV loops can occlude access to the adjacent bnAb epitopes and can be targeted by antibodies to elicit strain-specific responses rather than the more functional bnAb specificities they are shielding. Antibodies targeting these three epitopes accounted for more than half of the neutralizing antibodies in two natural infection cohorts: they corresponded to 57% of dominant specificities of the top 42 neutralizers in the IAVI Protocol C[22] and to 65% (33/51) of delineated epitope-specificities in the RV217 cohort[23]. Thus, modifying hypervariable loops to optimize access to adjacent bnAb epitopes on Env could improve HIV-1 Env vaccine candidates.

Previous studies analyzed HIV-1 antigens in which either V1, V2 or both loops were removed[24–26]. These variable loop deletions refocused immune responses, yet they also revealed that the removal of entire loops can lead to protein misfolding. Other studies used natural sequences with short HV loops, especially shorter V1HV, as candidate antigens. Zolla-Pazner and collaborators tested V1V2 from strain ZM53, which had a short V1, on scaffold proteins as vaccine antigens[27,28]. We selected a Circulating Recombinant Form (CRF) 01_AE strain with short V1HV and V2HV to be loaded on a self-assembling nanoparticle, V1V2-SHB-SAPN[29]. However, a natural sequence does not necessarily have ideal bnAb epitopes as well as ideal HV loops for eliciting bnAbs.

In this work, we systematically redesigned HV loops to optimize access to bnAb epitopes and overcome limitations associated with natural HIV-1 sequences. Our goals were to reduce the shielding of bnAb epitopes caused by HV loops and to limit the targeting of HV loops by strain-specific antibodies. Publicly available circulating subtype B and C and CRF01_AE sequences were analyzed to infer naturally occurring loop lengths and AlphaFold2[30] was used to predict protein structure to maintain structural integrity, loop anchoring, and the glycan shield. Env HV loops were redesigned for updated consensuses of subtypes B and C and CRF01_AE[31] that we recently created using publicly available[32] independent sequences sampled only since 2010. We used consensus sequences because they are a better representation of HIV-1 diversity than any natural sequence and are thus optimal as vaccine candidates[33–36]. We hypothesize that subtype consensus sequences with redesigned HV loops reduce the shielding of bnAb epitopes caused by HV loops limit the targeting of HV loops by strain-specific antibodies and show improved antibody binding for Env with redesigned HV loops.

## Results
### Short HV loops were rare for subtypes B, C, and CRF01_AE
We curated publicly available circulating HIV-1 sequences to obtain a dataset of 4847 independent Env sequences sampled from people living with HIV-1 (PLWH) between 1979 and 2020[31]. We focused on subtype B ($n = 2495$), subtype C ($n = 1503$) and CRF01_AE ($n = 849$) (Table S1) sequences since these subtypes/CRF are most frequently sequenced, thereby enabling robust analyses. We analyzed the distribution of HV loop lengths and the number of potential N-linked glycosylation sites (PNGS) in four HV loops (Fig. 1, Tables S1 and S2). We focused on the HV loops, which correspond to a portion of the variable loops: V1HV, sites 132-152 within sites 131-157 (V1); V2HV, sites 185-190 within sites 157-196 (V2); V4HV, sites 395-412 within sites 385-418 (V4); V5HV, sites 460-467 within sites 460-470 (V5). V3 was omitted because it does not contain an HV segment. The location of the HV loops, as well as adjacent bnAb epitopes, are shown on the structure in Fig. 1A.

V1HV loops showed a median length of 20 to 23 amino acids (AA), which was about twice as long as V2HV (7-9 AA), V4HV (9-12 AA), and V5HV (8-9 AA) loops (Fig. 1B, Table S1). The length distributions across subtypes/CRF were similar for V2HV and V5HV, while there were differences for V1HV and V4HV. CRF01_AE showed fewer short V1HV loops than subtype B or subtype C: there were 8% of CRF01_AE Env with V1HV ≤ 20 AA compared to 26% and 49% for B and C Env, respectively. Subtype B had fewer short V4HV loops than subtype C and CRF01_AE: there were 13% of subtype B Env with V4HV ≤ 10 AA compared to 49% and 65% for C and CRF01_AE Env, respectively. Similar patterns were observed for the number of PNGS on HV loops across subtypes/CRF (Fig. 1C, Table S2). Sequences with V1, V4, and V5 HV loops that were shorter than or equal to the 10th percentile were very rare representing 0.73%, 0.53%, and 0.32% of the dataset for subtypes B, C, and CRF01_AE, respectively. We also determined the loop lengths of sequences that have been used as inserts in previous vaccine efficacy trials or that are under consideration for future trials. We found that these insert sequences usually had one or more long HV loops (Table 1)[37–46]. For example, the subtype C TV1 sequence which was included in the vaccine tested in the HVTN702[45] vaccine efficacy trial had a V1HV loop that was 33 AA long while the median V1HV for subtype C is 20 AA. The insert Mosaic2.Env[40] used in the Mosaico and Imbokodo vaccine efficacy trials had a V2HV that was twice as long (16 AA) as the median for subtypes B and C (7 and 9 AA, respectively). We identified some exceptions with short HV loops: the subtype A1 92RW020[41] insert that was part of the trivalent vaccine tested in the HVTN505 trial and the subtype C CH505[44] insert that has been undergoing early clinical testing.

### Long HV loops associated with decreased neutralization sensitivity to bnAbs
The neutralization sensitivity to bnAbs was compared for HIV-1 Env with the shortest (≤5 percentile) and longest (≥95 percentile) HV loops. All neutralization and corresponding sequence data were obtained from CATNAP[47]. Six bnAbs belonging to three representative epitope groups were compared: V2-apex, PGT145 and VRC26.25; glycan supersite, 10-1074 and PGT135; and CD4bs, VRC01 and 3BNC117 (Fig. 2). To evaluate the effect of loop length on bnAb sensitivity, we compared short and long HV loops for the HV loop proximal to each bnAb (V2HV to V2-apex antibodies, V1HV to glycan supersite antibodies, and V5HV to CD4bs antibodies) and for the three other HV loops. The virus sequences with the longest adjacent HV loops had significantly higher half maximal inhibitory concentration (IC50) than the sequences with the shortest HV loops ($p$-value ≤ 0.015, one-tailed Mann–Whitney U test) for the six bnAbs (Fig. 2), suggesting that longer HV loops obstruct bnAb epitope availability. When HV loops distant from the bnAb epitope (V5HV to V2-apex and glycan supersite antibodies and V2HV to CD4bs antibodies) were compared, there was no significant difference in sensitivity for short or long HV loops for five of the six bnAbs tested ($p$-value ≥ 0.207). Interestingly, there might be some longer-range interactions between HV loops. Sequences with long/short V1HV had a similar impact on IC50 values for V2-apex antibodies as V2HV, but there is no association between the length of V1HV and V2HV (Figure. S1), suggesting that interactions between V1HV and V2HV may also modulate the accessibility of V2-apex epitopes. For VRC01, sequences with shorter HV loops for V1, V2, and V5 were significantly associated with increased sensitivity when compared to sequences with longer HV loops (Fig. 2E). The strongest impact was seen for V1HV loops and the neutralization sensitivity to 10-1074. HIV-1 Envs with the shortest V1HV loops had a median IC50 greater than or equal to 5000 times lower than the strains with the longest V1HV loops (0.02 µg/ml vs. ≥ 100 µg/ml, one-tailed Mann-Whitney U test $p$-value = 0.00003). For V5HV loops, which are distant

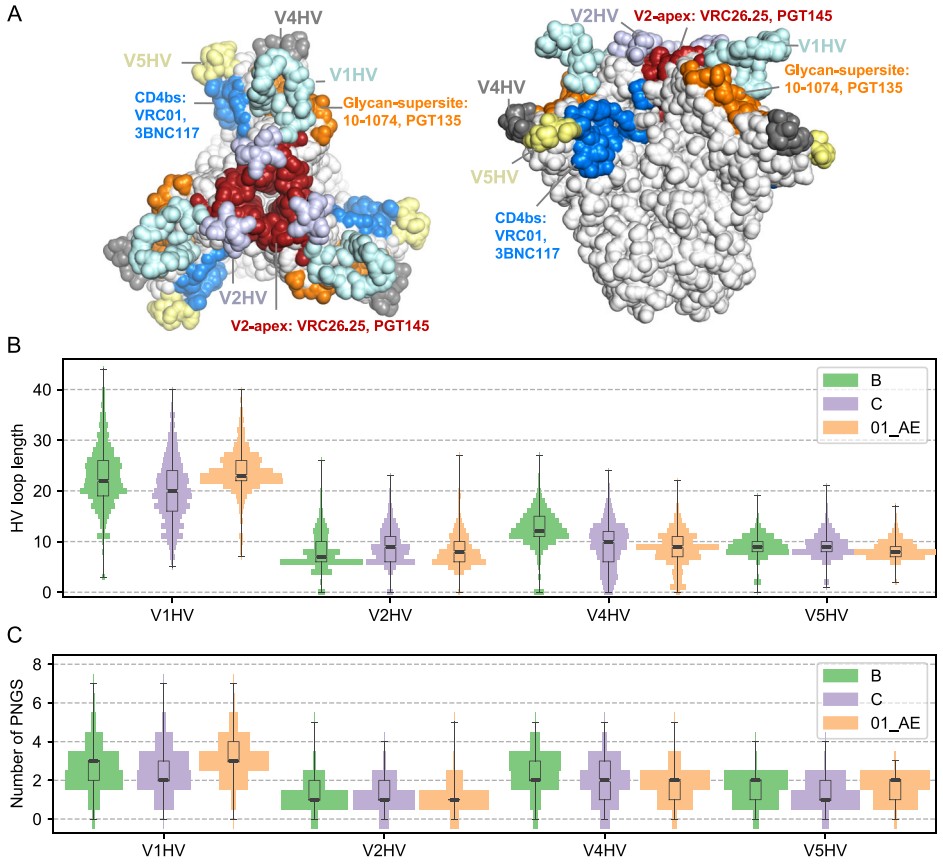

**Fig. 1 | Characteristics of hypervariable (HV) loops. A** Location of HV loops and related HIV-1 bnAb epitopes on Env prefusion structure. **B** Distribution of lengths for V1, V2, V4, and V5 HV loops and **C** distribution of the number of potential N-linked glycosylation sites (PNGS) for subtypes B ($n = 2495$) and C ($n = 1503$) and CRF01_AE ($n = 849$) Env sequences. The width corresponds to the proportion of sequences in the subtype/CRF sample of sequences and is normalized for each HV loop. The box plots depict the median and 25th/75th percentiles of the distribution, with the minima and maxima shown as whiskers. Source data for panels **B** and **C** are provided in the Source Data files.

from the 10-1074 epitope, there was no significant difference in sensitivity when short and long loops were compared (one-tailed Mann-Whitney U test $p$-value = 0.838). Importantly, these data masked some subtype-specific differences, as differences between short and long HV loops were not seen for subtype B (with $p$-values ≥ 0.139) except for 10-1074 ($p = 3.6e{-}07$) (Figure. S2). Differences for subtype C were generally statistically significant, except for 10-1074 ($p = 0.129$) and PGT135 ($p = 0.128$). We repeated the analysis using 10−25% as the threshold to compare the extremes of the distributions (Figure. S3). Differences between short/long adjacent HV loops remained statistically significant for most bnAbs ($p < 0.045$, Figure. S3C-F) or borderline significant for VRC26.25 when the cutoff was set as 25% ($p = 0.079$, Figure. S3B), with PGT145 as an exception ($p ≥ 0.101$ with 10-25% as the threshold, Figure. S3A). Overall, these results indicated that a long HV loop can diminish the virus's sensitivity to a bnAb.

**Strategy to redesign HV loops**

We redesigned the HV loops on Env to improve accessibility to typical bnAb epitopes. We used Env consensus sequences which were derived only from sequences that were sampled after 2010, thus reflecting currently circulating viruses[31]. Our goals were to: (1) make the loop as short as possible while maintaining structural integrity; (2) preserve the loop anchor sites, which are the relatively conserved start and end of the HV loops that form favorable interactions with the non-HV part of Env; and (3) maintain the glycan shield by ensuring there was at least

one PNGS on each redesigned loop. Using the V1HV loop of subtype B as an example, we analyzed the alignment of circulating sequences (Fig. 3A) and the AlphaFold2 predicted structure of the corresponding consensus sequence (Fig. 3B). We identified the relatively conserved TDL motif (sites 132-134) as N-anchor sites and the relatively conserved MEKG motif (sites 149-152) as C-anchor sites, with all sites in between considered spacer sites. In the TDL motif, D133 may form favorable charge-charge interactions with K151 in the C-anchor sites or with K155; L134 may form hydrophobic interactions with I154, L175, I323, and I326. In the MEKG motif, M149 may form hydrophobic interactions with I326 and I154; E150 may form a favorable charge-charge interaction with R419. These anchor sites were retained since they did not have direct contact with glycan supersite antibodies. To shorten the spacer, we evaluated different lengths iteratively using AlphaFold2 predicted structures by comparing the predicted local Difference Distance Test (lDDT) scores, which measure the confidence given by AlphaFold2 for each output structure. A specific spacer was selected when it maintained or increased the predicted lDDT for the anchor and surrounding sites of the HV loop, suggesting that the anchor stability will be maintained. For V1HV, the spacer was shortened from 15 AA to 5 AAs. This resulted in the V1HV loop length changing from 22 AA (50th percentile in subtype B sequences) to 12 AA (4th percentile in subtype B sequences) (Fig. 3B, C). The V1HV, V2HV, and V5HV loops of subtypes B, C, and CRF01_AE were redesigned with the same strategy and are shown in Table 2 and Figure. S4–S11.

**Table 1 | Comparison of redesigned HV loops to those found in vaccine candidates**

| Sequence | Subtype/CRF | Efficacy trials | V1HV | | V2HV | | V4HV | | V5HV | |
|---|---|---|---|---|---|---|---|---|---|---|
| | | | $N_{AA}$[a] | $N_{PNGS}$ | $N_{AA}$[a] | $N_{PNGS}$ | $N_{AA}$[a] | $N_{PNGS}$ | $N_{AA}$[a] | $N_{PNGS}$ |
| A244 | CRF01_AE | VAX003, RV144 | 28 (90%) | 5 | 8 (62%) | 1 | 9 (65%) | 1 | 9 (75%) | 2 |
| MN | B | VAX003, VAX004, RV144 | 27 (81%) | 3 | 6 (45%) | 1 | 12 (52%) | 2 | 10 (78%) | 1 |
| GNE8 | B | VAX004 | 20 (36%) | 2 | 6 (45%) | 1 | 15 (83%) | 3 | 7 (12%) | 1 |
| 92TH023 | CRF01_AE | RV144 | 23 (52%) | 3 | 8 (62%) | 1 | 9 (65%) | 2 | 7 (27%) | 2 |
| 92RW020 | A1 | HVTN505 | 9 (2%) | 2 | 8 (61%) | 1 | 12 (67%) | 3 | 7 (16%) | 1 |
| HXB2 | B | HVTN505 | 21 (43%) | 2 | 6 (45%) | 1 | 15 (83%) | 2 | 8 (36%) | 1 |
| 97ZA012 | C | HVTN505 | 15 (23%) | 2 | 11 (76%) | 2 | 2 (9%) | 0 | 8 (44%) | 0 |
| 96ZM651 | C | HVTN702 | 26 (86%) | 4 | 14 (93%) | 2 | 6 (27%) | 1 | 12 (92%) | 2 |
| TV1 | C | HVTN702 | 33 (98%) | 4 | 6 (25%) | 0 | 13 (87%) | 2 | 9 (60%) | 2 |
| 1086 | C | HVTN702 | 12 (12%) | 1 | 9 (62%) | 1 | 12 (79%) | 3 | 10 (76%) | 2 |
| Mosaic1.Env | – | HVTN705, HVTN706 | 25 (74%) | 3 | 8 (61%) | 2 | 16 (93%) | 3 | 8 (43%) | 1 |
| Mosaic2.Env | – | HVTN705, HVTN706 | 21 (45%) | 3 | 16 (97%) | 2 | 9 (33%) | 2 | 11 (89%) | 1 |
| CH505 | C | – | 12 (12%) | 2 | 6 (25%) | 1 | 15 (94%) | 3 | 6 (6%) | 1 |
| WITO | B | – | 19 (26%) | 3 | 8 (68%) | 1 | 11 (37%) | 2 | 8 (36%) | 2 |
| DU422 | C | – | 21 (62%) | 3 | 10 (70%) | 1 | 9 (49%) | 2 | 7 (18%) | 1 |
| BG505 | A1 | – | 13 (8%) | 2 | 14 (94%) | 2 | 14 (83%) | 3 | 8 (43%) | 1 |
| JR-FL | B | - | 19 (26%) | 3 | 6 (45%) | 1 | 11 (37%) | 2 | 8 (36%) | 1 |
| B_2010s[b] | B | - | 12 (4%) | 1 | 4 (11%) | 1 | 11 (37%) | 3 | 7 (12%) | 1 |
| C_2010s[b] | C | - | 13 (16%) | 1 | 5 (8%) | 1 | 11 (69%) | 3 | 7 (18%) | 1 |
| 01_AE_2010s.1[b] | CRF01_AE | - | 9 (1%) | 1 | 5 (12%) | 1 | 11 (84%) | 3 | 7 (27%) | 2 |
| 01_AE_2010s.2[b] | CRF01_AE | - | 19 (8%) | 1 | 5 (12%) | 1 | 11 (84%) | 3 | 7 (27%) | 2 |

[a]HV loop length with the percentile for this loop length in corresponding subtypes shown in parenthesis, except for Mosaic1.Env, Mosaic2.Env, and subtype A1 sequences for which a pooled set of subtype B, C, and CRF01_AE sequences was used to calculate the percentile for the specific loop length.
[b]Consensus sequences with redesigned HV loops. 01_AE_2010s.2 includes the 'IGNITD' segment in the V1HV and is 10 AA longer in V1HV than 01_AE_2010s.1.

Contrary to the strategy for V1HV, V2HV, and V5HV, for V4HV loops we did not seek to obtain the shortest V4HV loops in order to accommodate its specific structural context. Sites adjacent to V4HV (non-HV sites within 10 Å of the HV loops, relative surface area > 0.30) were diverse, and no major bnAbs are known to target this segment of Env. For subtype C, the median Shannon entropy of V4HV adjacent sites was 2.4 bits compared to 0.8, 0.7 and 1.1 bits, respectively, for the sites adjacent to V1HV, V2HV and V5HV (Fig. 4A). This high Shannon entropy for V4HV loops indicated that there were 4.5 ($2^{2.4/1.1}$) to 10.8 ($2^{2.4/0.7}$) times more AA possibilities on average for surface sites adjacent to V4HV compared to the other HV loops. In addition, the distance encompassing spacer sites was larger for V4HV than for V1HV, V2HV, and V5HV (20.8 Å for V4HV versus 10.8, 7.9, and 9.4 Å for V1HV, V2HV, and V5HV, respectively, based on the AlphaFold2 model of the subtype B consensus). This suggests that a short V4HV spacer may not be compatible with the structure. Hence, we chose a medium length for the V4HV loop; this loop length could also partially shield the highly diverse segment adjacent to the V4HV. For the subtype B consensus, we identified W395 as the N-terminal anchor site, as it is well conserved and forms hydrophobic interaction with nearby non-HV sites. We also assigned ND (sites 411–412) as the C-terminal anchor sites, as N is the most frequent option and D/E412 might form a charge-charge interaction with R335. The spacer was replaced with an 11-AA linker (Fig. 4B–D). The redesigned linker contained two PNGS, which was the median number for V4HV of subtypes B and C, and CRF01_AE sequences (Fig. 1C). The V4HV loops of subtype C and CRF01_AE were redesigned with the same strategy (Figure. S12, S13). One notable feature of the V4HV redesign for CRF01_AE is that C395 (which was predicted to form a disulfide bond in our sequence) was replaced with the second most frequent residue at that site, W, and the spacer was extended to better shield the surrounding sites of V4HV (Figure. S13). Across all redesigned HV loops, the spacer consisted mainly of glycine

and serine which lack the side chains that are preferentially targeted by antibodies and may also increase the flexibility of the HV loops. The redesigned HV loops tended to have higher predicted lDDT scores from AlphaFold2 than the unmodified ones, except for the V4HV of the subtype C consensus (Figure. S14). All redesigned HV loops are listed in Table 2.

**Updated Env consensus sequences are functional Env proteins**
We produced pseudoviruses that expressed Env proteins for subtypes B, C, and CRF01_AE with either modified or unmodified loops. We showed that the seven Env tested were infectious. When normalized by the Gag-p24 quantities, viral titers were lower for the pseudoviruses with redesigned Env loops compared to those with unmodified loops for subtypes B and C, while it was the opposite for CRF01_AE (Fig. 5A).

Next, we produced our consensus sequences as gp140 glycoproteins. Negative-stain electron-microscopy (EM) showed that gp140s for subtypes B, C, and CRF01_AE were mostly open trimers and monomers, with only monomers being averageable from negative stain EM (Figure. S15).

Finally, we stabilized Env in the prefusion conformation by combining previously reported modifications[48,49] and three extra mutations. As stabilized trimers, produced by transient transfection and purified by GNA-lectin and size-exclusion chromatography, consensus B and C Env were just under 50% trimer by 2D averaging of negative-stain EM data (42 and 45%, respectively); the HV loop redesign marginally increased the population of trimer observed to 59 and 55%, respectively (Fig. 5B). To determine whether the HV loop redesign affected the opening of the trimer, we used 3D classification and refinement without any applied symmetry. For the subtype B consensus, 76% of trimers classified with the reconstruction identifiable as the closed prefusion state, while 24% appeared open, but still in a prefusion state (Fig. 5C). Results were similar for the HV loop redesign

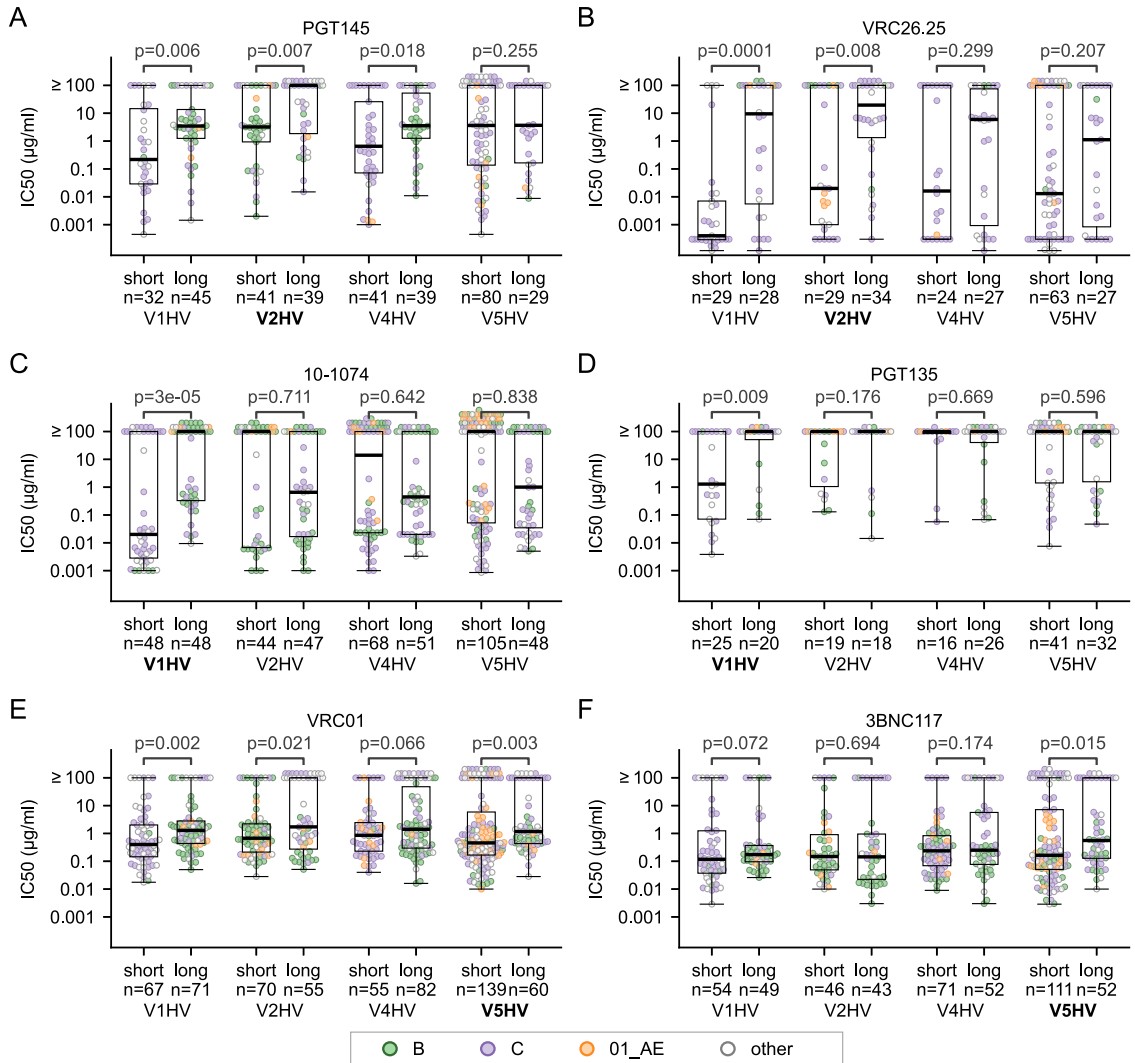

**Fig. 2 | Comparison of the bnAb sensitivity for HIV-1 Env with short (≤ 5 percentile) or long (≥ 95 percentile) hypervariable (HV) loops.** For each panel, comparisons of IC50 values for short and long HV loops are shown for representative antibodies: PGT145 (**A**) and VRC26.25 (**B**), two V2 apex-targeting antibodies whose epitopes are close to V2HV loops; 10-1074 (**C**) and the PGT135 (**D**), two glycan supersite antibodies whose epitopes are close to V1HV loops; VRC01 (**E**) and 3BNC117 (**F**), two CD4 binding site antibodies with epitopes close to the V5HV loops. The HV loop nearest to the epitope of each antibody is shown in bold font. The box plots depicted the median and 25th/75th percentiles of the distribution, with the minima and maxima as whiskers. The *p*-values from non-paired, non-parametric one-tailed Mann-Whitney U tests are provided. IC50 values equal to or greater than 100 µg/ml are grouped (upper limit of antibody neutralization measurements in the CATNAP database). Source data for all panels are provided in the Source Data files.

with 68% in the closed and 32% in the open prefusion state. In contrast, the subtype C consensus trimers were only at 31% in the closed prefusion state with 69% appearing open, with minimal changes with the HV loop redesign (29% in the closed and 71% in the open prefusion state). We cannot rule out that the conformational landscape of stabilized trimers is affected by the protein production and purification process as well as the adsorption to the carbon grid, which may explain the substantial fraction of monomer observed by negative-stain EM. Overall, these results suggest that the HV loop redesign marginally increased the trimer yield and had limited impact on the conformation of Env.

### Redesigned HV loops improved accessibility to bnAb epitopes in silico

To measure in silico if the redesign improved exposure of bnAb epitopes, we calculated a depth index that corresponds to the closest distance between an epitope site and the antibody probe. The antibody probe is a sphere with a radius corresponding to an antibody's Fv (=30 Å) that rolls around the Env structure (Fig. 6A). A short depth indicates that the site is more easily accessible to the antibody. We calculated the depth corresponding to the epitopes of two glycan supersite bnAbs, 10-1074 and PGT135, and two CD4bs bnAbs, VRC01 and 3BNC117, before and after the HV loop redesign. We remodeled each loop 500 times via the kinematic closure (KIC) loop modeling protocol[50] from PyRosetta[51]. Among the 500 models, the 100 models with the lowest Rosetta energy were selected for the epitope depth calculation. The epitope depth decreased for the four bnAb epitopes of the Env consensus of subtypes B, C, and CRF01 AE (Fig. 6B–D). The median depth decreased by 0.8 to 1.6 Å for glycan supersite antibody epitope, while the median depth decreased by 0.2 to 0.8 Å for CD4bs antibody epitope. Besides the depth, the clashes between HV loops and corresponding antibodies were also calculated to show the impact of the HV loop redesign on the antibody accessibility of bnAb epitopes. The redesigned Env alleviated the clashes between 10-1074 and PGT135 and V1HV, but with limited effect on the clashes between CD4bs antibodies and V5HV (Fig. 6E, F). The greater depth and clash reduction for glycan-supersite epitope sites near V1HV was likely due to the fact that V1HV loops were longer than other HV loops and the HV loop

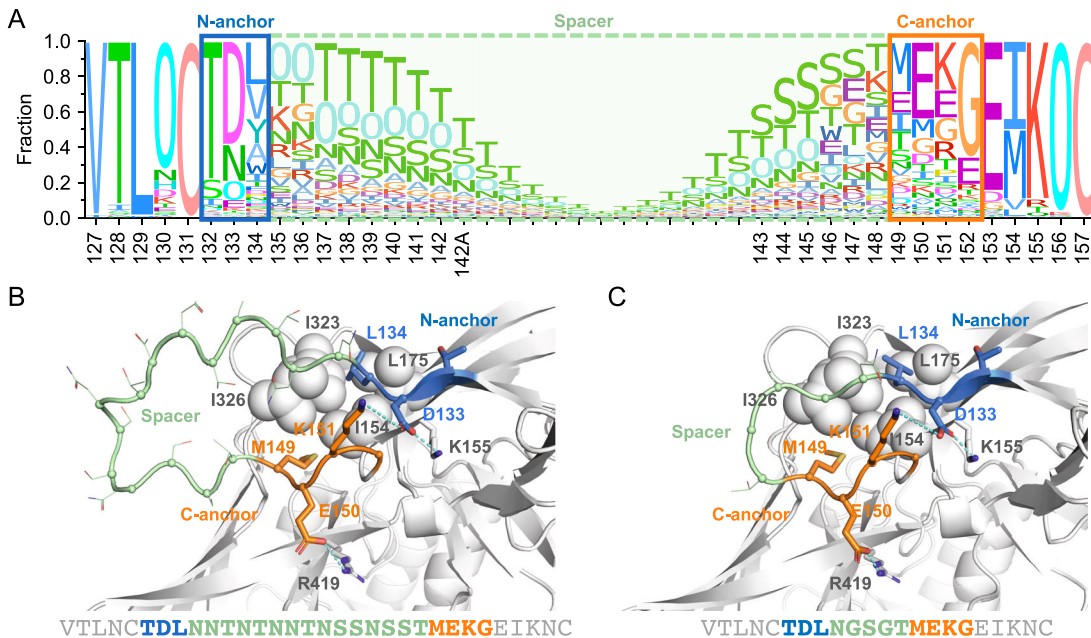

**Fig. 3 | Redesigned V1 hypervariable (HV) loop of the subtype B Env consensus sequence. A** Sequence logo of 2495 subtype B sequences around the V1HV loop, with the N-/C-anchors and spacer indicated by boxes. The height of each amino acid letter represents its frequency at the position. The letter 'O' indicates a potential N-linked glycosylation site. The consensus B Env with unmodified V1HV loop (**B**) and redesigned V1HV loop (**C**) modeled by AlphaFold2 are shown with the N-anchor, C-anchor, and spacer sites colored blue, orange, and light green, respectively. Non-HV residues that interact with anchor sites are shown as spheres (hydrophobic interactions) or sticks (charge-charge interactions). Source data for panel **A** are provided in the Source Data file.

## Table 2 | Sequences of redesigned HV loops

| Variable loop | Subtype/CRF | Original sequence | Redesigned |
|---|---|---|---|
| V1HV | B | TDLNNTNTNNTNSSNSSTMEKG | TDLNGSGTMEKG |
| V1HV | C | TNVNNTNTNTNNNTNNNIKE | TNVNGSGNDTMKE |
| V1HV | CRF01_AE | TNANLTNTNTTNNTNTTNIGNITD | TNVNGSGGD (1)<br>TNANLTGNGSGGSIGNITD (2) |
| V2HV | B | DNNNNTS | DNTS |
| V2HV | C | DNNNNSSE | NNGSE |
| V2HV | CRF01_AE | DNNNSSE | GNSSE |
| V4HV | B | WNNNSTTNNNTNTNDT | WNGNGSGNGSGSNDT |
| V4HV | C | YNGNGNNNNTNSNST | YNGNGSGNGSGSNST |
| V4HV | CRF01_AE | CIGNETTTEGCNGT | WNGNGSGNGSGSNGT |
| V5HV | B | NNNTNTTET | NNGTTET |
| V5HV | C | NNNNNTTET | NNGTTET |
| V5HV | CRF01_AE | NNNNTNET | NNGTNET |

redesign removed more spacer residues than for other HV loops (Fig. 1A, Table 2). The apex antibodies, PGT145 and VRC26.25, were not tested because their epitopes are quaternary and require modeling of the entire Env trimer, which is currently beyond our computational power.

We further examined the modeled structure of CRF01_AE V1HV because CRF01_AE Env has longer V1HV than those of subtypes B and C (Fig. 1B) and shows high resistance to glycan supersite antibodies[52]. The consensus sequence had a 'IGNITD' motif (147-152) as its V1HV C-anchor sites (Figure. S5AB). AlphaFold2 predicted that I147 occupied the hydrophobic pocket formed by R327, K328, Y330 and P417, which corresponds to the site targeted by F[100G] on the CDR-H3 of 10-1074 (Fig. 6E). After the HV loops redesign, the 24 AA-long V1HV loop (58.2 percentile in CRF01_AE) was replaced with a 9 AA-long loop (0.6 percentile in CRF01_AE, 01_AE.1) (Figure. S5C), hence, the median depth of 10-1074 epitope sites was reduced from 6.3 to 4.8 Å. The redesigned Env reduced the average number of clashes with 10-1074 from 120.8 to

9.2 atoms (Fig. 6E). A similar decrease in the number of clashes was observed for another glycan supersite bnAb, PGT135, but the impact was limited for CD4bs antibodies (VRC01 and 3BNC117) (Fig. 6F). Because the 'IGNITD' motif is highly conserved in CRF01_AE viruses, another version of V1HV (01_AE.2) was designed to preserve the 'IGNITD' motif, which may be important for CRF01_AE specific bnAbs (Figure. S5D). This variant CRF01_AE.2 design also improved antibody accessibility for glycan super site antibodies albeit to a lesser extent (Fig. 6F and S16).

### Improved antibody binding for Env-gp140s with redesigned HV loops

To test if the HV loop redesign improves antibody binding to Env, we analyzed binding to the Env-gp140 glycoproteins. Both versions of the consensus subtype B and C, plus the unmodified CRF01_AE consensus and the redesigned V1HV with the conserved 'IGNITD' motif (01_AE.2) were tested for binding to plasma pools and monoclonal antibodies

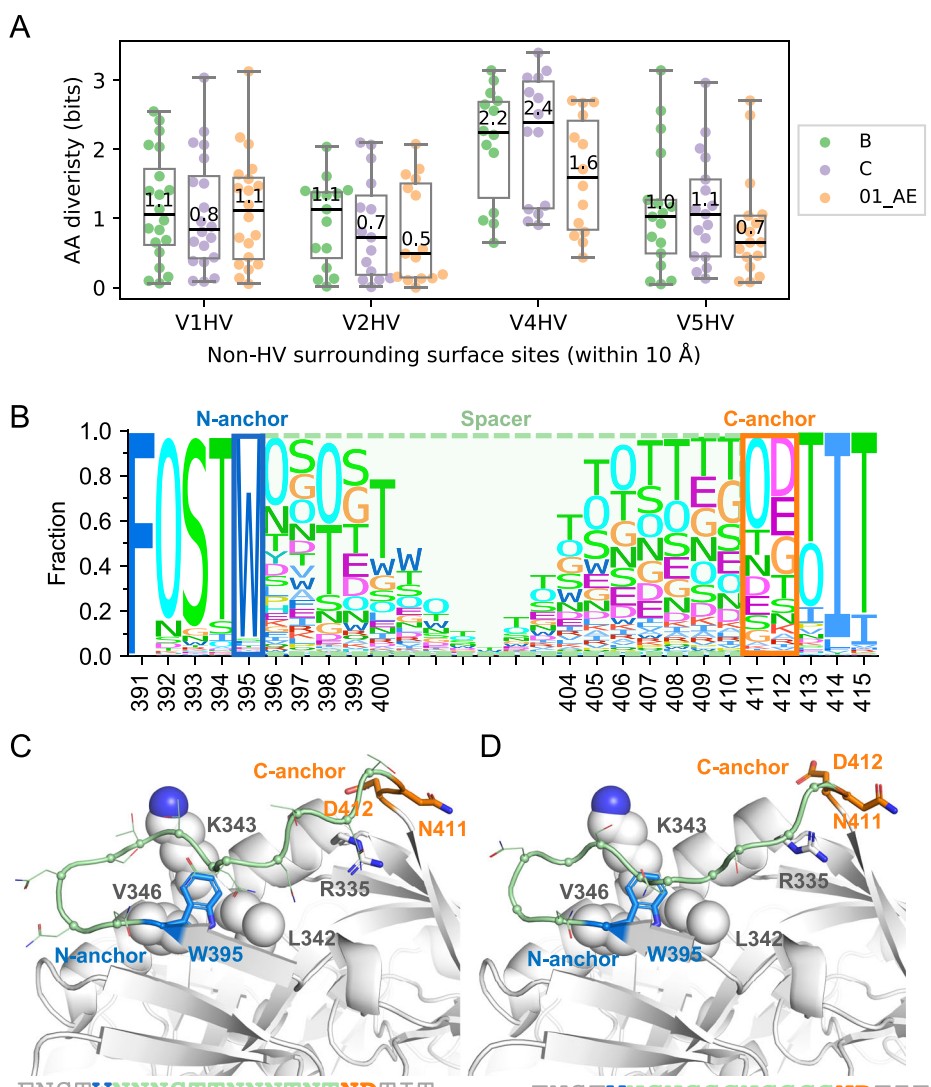

**Fig. 4 | Redesigned V4 hypervariable (HV) loop of the subtype B Env consensus sequence.** **A** The Shannon entropy of non-HV surrounding surface sites is shown for V1HV ($n = 20$), V2HV ($n = 15$), V4HV ($n = 14$) and V5HV ($n = 17$) of each subtype/CRF, with median values reported. The box plots depict the median and 25th/75th percentiles of the distribution, with the minima and maxima shown as whiskers. (**B**) Sequence logo of 2495 subtype B viruses around the V4HV loop, with the N-/C-anchor and spacer labeled. The height of each amino acid letter represents its frequency at the position. The letter 'O' represents potential N-linked glycosylation sites. The consensus B Env with unmodified V4HV (**C**) and redesigned V4HV loop (**D**) modeled by AlphaFold2 are shown with the N-anchor, C-anchor, and spacer sites colored blue, orange, and light green, respectively. Non-HV residues that interact with anchor sites are shown as spheres (hydrophobic interactions) or sticks (charge-charge interactions). Source data for panels (**A**) and (**B**) are provided in the Source Data files.

with a bead-based multiplex immunoassay[53,54]]. The pooled plasma samples represented 13 cohorts of PLWH spanning diverse geographic areas and subtype/CRF distributions. Plasma antibodies across multiple clade-specific pools recognized the consensus B Env with redesigned HV loops at a slightly higher magnitude of binding compared to the Env with unmodified HV loops (Fig. 7A). Similarly, monoclonal antibodies recognized the subtype B Env-gp140s with redesigned loops at a higher magnitude of binding, especially for antibodies targeting the CD4bs, V2 apex and glycan supersite epitopes (Fig. 7B). Summary measures for binding of pooled plasma or monoclonal antibodies demonstrated minor but significantly higher recognition of the subtype B Env-gp140s with redesigned HV loops when compared to the Env with unmodified loops ($p \le 0.008$) (Fig. 7C, D). A similar improvement of antibody binding was seen for the CRF01_AE consensus with redesigned HV loops ($p = 0.008$) (Fig. 7C, D, Figure. S17). The subtype C consensus proteins with or without redesigned loops showed no difference in antibody binding (Fig. 7C, D, Figure. S17), consistent with consensus C Env exhibiting high binding to plasma

relative to the other subtype/CRF before the loops were redesigned. Binding to monoclonal antibodies that target epitopes that are not presented on gp140 such as MPER antibodies and PGDM1400 (which targets the quaternary V2-apex epitope) was low as expected and was excluded from the statistical test. We compared the change in the predicted epitope depth to the change in antibody binding signal and found no significant correlation between them (Spearman's $\rho = 0.52$, $p$-value = 0.15) (Figure. S18). In addition, we transfected 293 T cells with Env-gp160 plasmids corresponding to unmodified and redesigned HV loops and evaluated bnAb binding. For the two subtype B consensus sequences, we showed comparable Env cell surface expression, and staining with bnAbs such as 3BNC117 and PGDM1400 indicated that Env were expressed as closed trimers. Moreover, seven bnAbs tested showed increased binding to the Env with redesigned HV loops, suggesting improved epitope display (Figure. S19).

Finally, we evaluated how the Env consensus proteins (with or without modified HV loops) were neutralized by bnAbs. We used a pseudovirus neutralization assay in TZM-bl cells and tested ten bnAbs

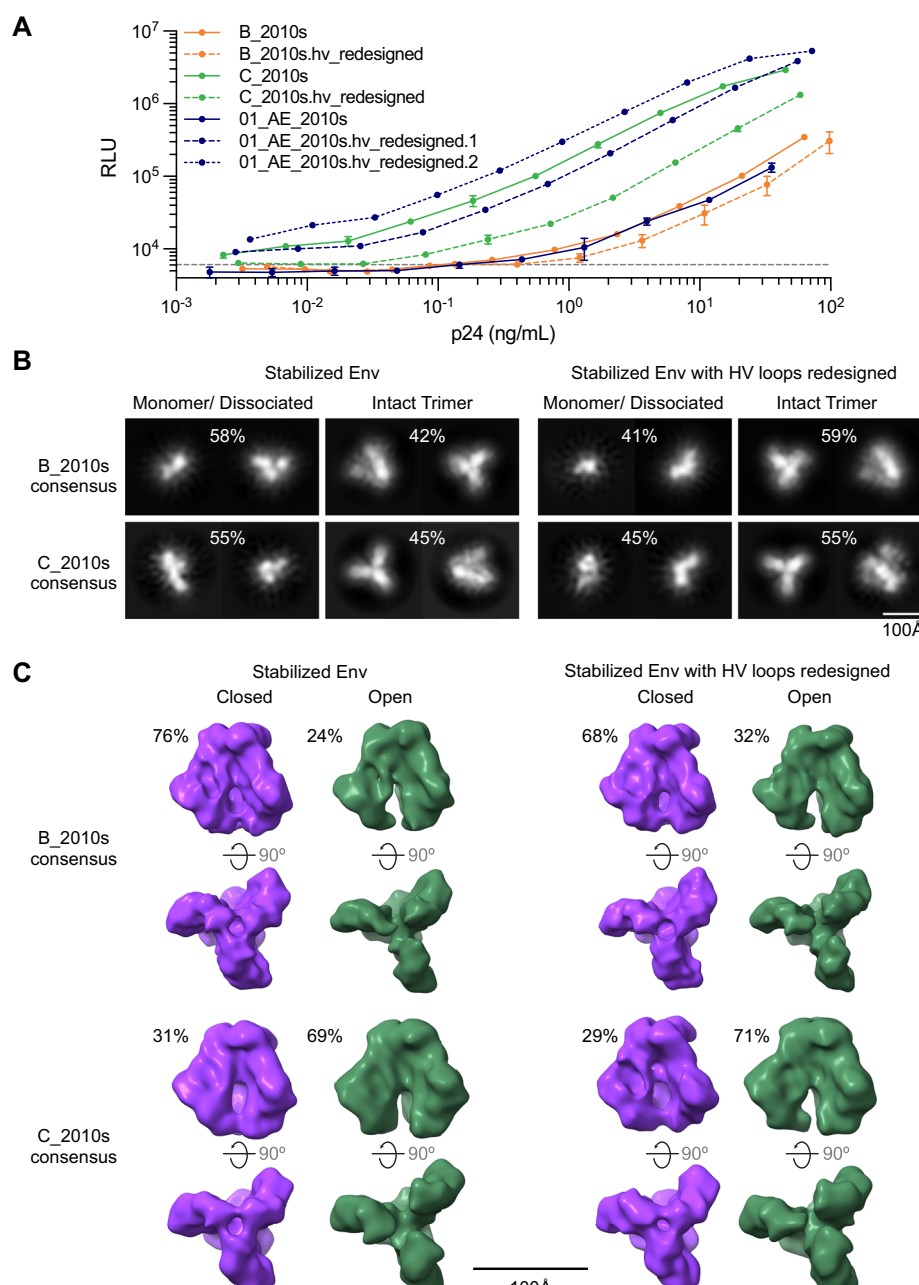

**Fig. 5 | Updated Env consensus proteins for subtypes B, C, and CRF01_AE are functional. A** Viral titers for Env consensus proteins with or without redesigned hypervariable (HV) loops are presented as mean +/- SEM of three replicates. **B** Negative-stain electron microscopy for stabilized trimers with and without redesigned loops for consensus subtypes B and C. Representative 2D class averages of putative monomeric or disassembled Env and intact trimers for stabilized trimers with and without redesigned HV loops corresponding to the consensus for subtypes B and C. Percentages above the images indicate the relative proportion of particles between monomer and trimer classes. **C** 3D reconstructions of particles classified as trimers in 2D with no symmetry applied (C1). Classes resembling the closed, prefusion conformation of HIV-1 Env are shown in purple while classes resembling the open, prefusion conformation are shown in green. Percentages above each map indicate the relative proportion between closed and open prefusion conformations. A side view for each reconstruction with the trimer apex on the bottom (top) and the trimer apex facing the viewer (bottom) is shown for each reconstruction. Source data of **A** are provided in the Source Data file.

targeting the CD4 binding site (VRC01 and 3BNC117), the V2-apex (PG9 and PGT145), the glycan supersite (10-1074 and PGT128), the MPER (10E8 and 4E10) and the carbohydrate-specific mannose-dependent (2G12) epitopes as well as the soluble CD4 (sCD4) (Fig. 7E, G for IC50 values and S20 for IC80 values). When comparing the consensus Env with the redesigned HV loop Env to those with the native HV loops, the neutralization sensitivity did not increase or decrease consistently for all the bnAbs (Fig. 7E, G and S20). When pooling data across the 10 bnAbs, there was no significant increase in neutralization sensitivity for subtype B and CRF01-AE with redesigned HV loops (one-tailed exact Wilcoxon-Pratt signed-rank test, $p > 0.201$) (Fig. 7F). In contrast, the subtype C Env with redesigned HV loops was more sensitive to the ten bnAbs than the subtype C Env with unmodified loops ($p = 0.047$ for IC50 values (Fig. 7F) and $p = 0.016$ for IC80 values (Figure. S20B)). One bnAb, PG9, showed a notable increase in sensitivity for the antigens with redesigned HV loops with a similar pattern across subtypes/CRF (Fig. 7G). Moreover, CRF01_AE consensus proteins with the redesigned HV loop version 1 became sensitive to the glycan supersite bnAb 10-1074 (IC50 = 0.544 µg/ml) and to the sCD4 (IC50 = 0.617 µg/ml) while the original CRF01_AE consensus was resistant (IC50/80 > 50 µg/ml)

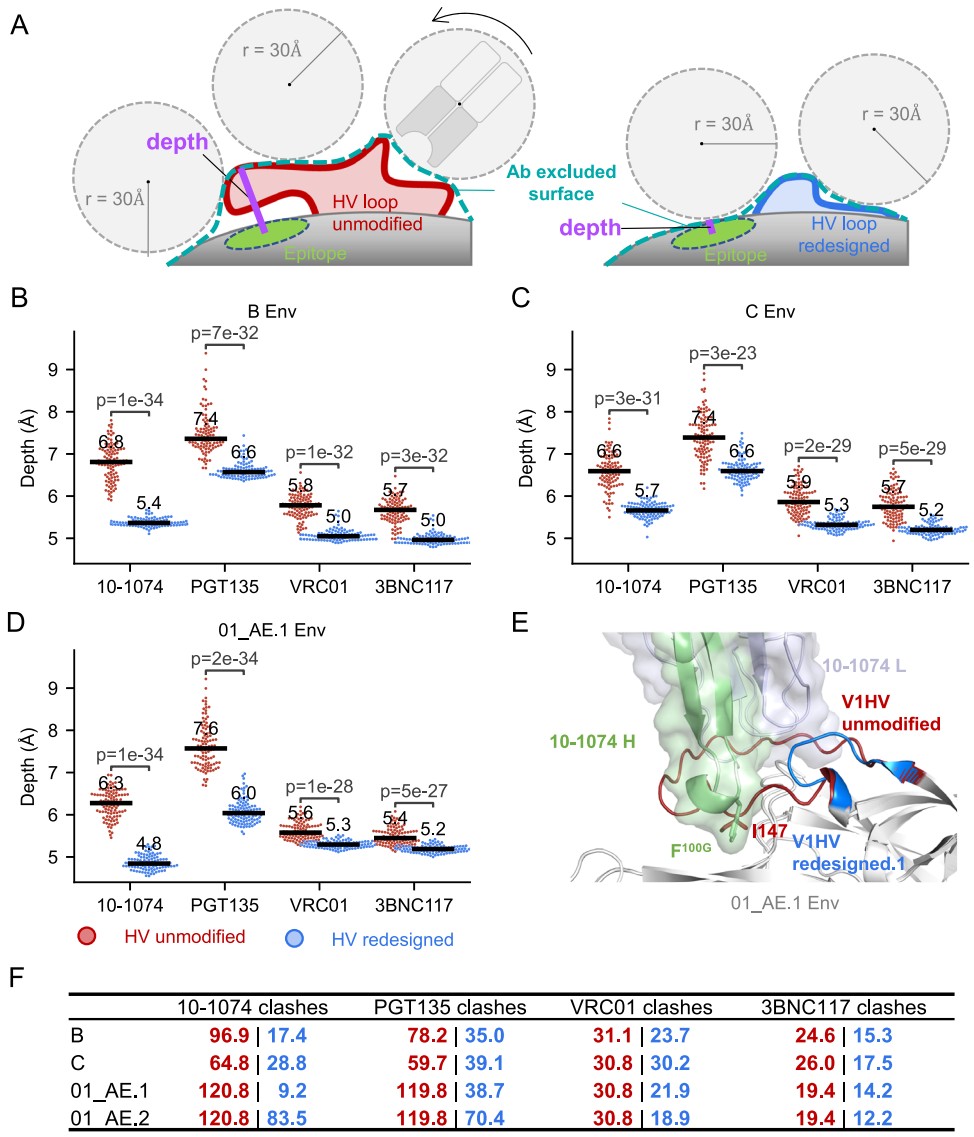

**Fig. 6 | Impact of redesigned hypervariable (HV) loops on the antibody accessibility of epitope sites. A** Schematic showing how the minimum distance between an epitope site and the sphere probe (radius 30 Å) representing an antibody, defined as depth, is used to detect antibody accessibility differences of Env surface sites before (red) and after (blue) the HV loop redesign. **B–D** Comparison of the depth for four representative antibodies before and after the HV loop redesign. Median depth is labeled. The *p*-value from non-paired, non-parametric one-tailed Mann-Whitney U tests is indicated (*n* = 100 lowest energy conformations selected from 500 modeled conformations). **E** Consensus CRF01_AE Env with unmodified and redesigned HV loops (AlphaFold2 prediction) are superimposed with Env in the 10-1074-Env complex. **F** Average decrease in clashes between antibody atoms and Env for unmodified (red) and redesigned (blue) cognate HV loops (based on 100 conformations). Source data for panels **B–D** are provided in the Source Data file.

(Fig. 7E, G, S20). Interestingly, the CRF01_AE consensus with redesigned HV loop version 1 was sensitive to 10-1074 while the Env with version 2 of the redesign was resistant, in agreement with the antibody access predictions described above. The redesign version 1 is shorter in V1HV indicating that resistance to 10-1074 in CRF01_AE is associated with the V1 loop length and is not necessarily mediated by the lack of glycosylation at site 332 as neither redesigned consensus Env had a glycan at site 332.

## Discussion

Here we redesigned the hypervariable loops V1HV, V2HV, V4HV, and V5HV using a structural modeling strategy to optimize access to critical bnAb epitopes. The redesign was done for subtypes B and C and CRF01_AE. These subtypes/CRF were selected because large datasets of sequences were available (for the next most frequent subtype, A1, only 288 sequences were available). In addition, HIV-1

subtype B is the most thoroughly characterized subtype, subtype C is responsible for the majority of cases among PLWH, and CRF01_AE is the virus that was circulating in the only HIV-1 vaccine trial where some efficacy was recorded[39]. We redesigned HV loops in Env consensus sequences that we had recently created; these consensus sequences reflected currently circulating viruses since they were derived only from sequences that were sampled after 2010[31]. The LANL database recently updated HIV-1 consensus sequences using all sequences sampled since the beginning of the epidemic[55], hence, these 2021 consensus sequences tend to be biased towards past sequences which are more numerous than sequences sampled in the past decade. We chose to restrict our consensus sequences to sequences sampled since 2010 to ensure an accurate representation of recently circulating viruses. We note that HIV-1 proteins that are available at the HIV Reagent Program (hivreagentprogram.org) and are commonly used as reagents in experimental assays were often

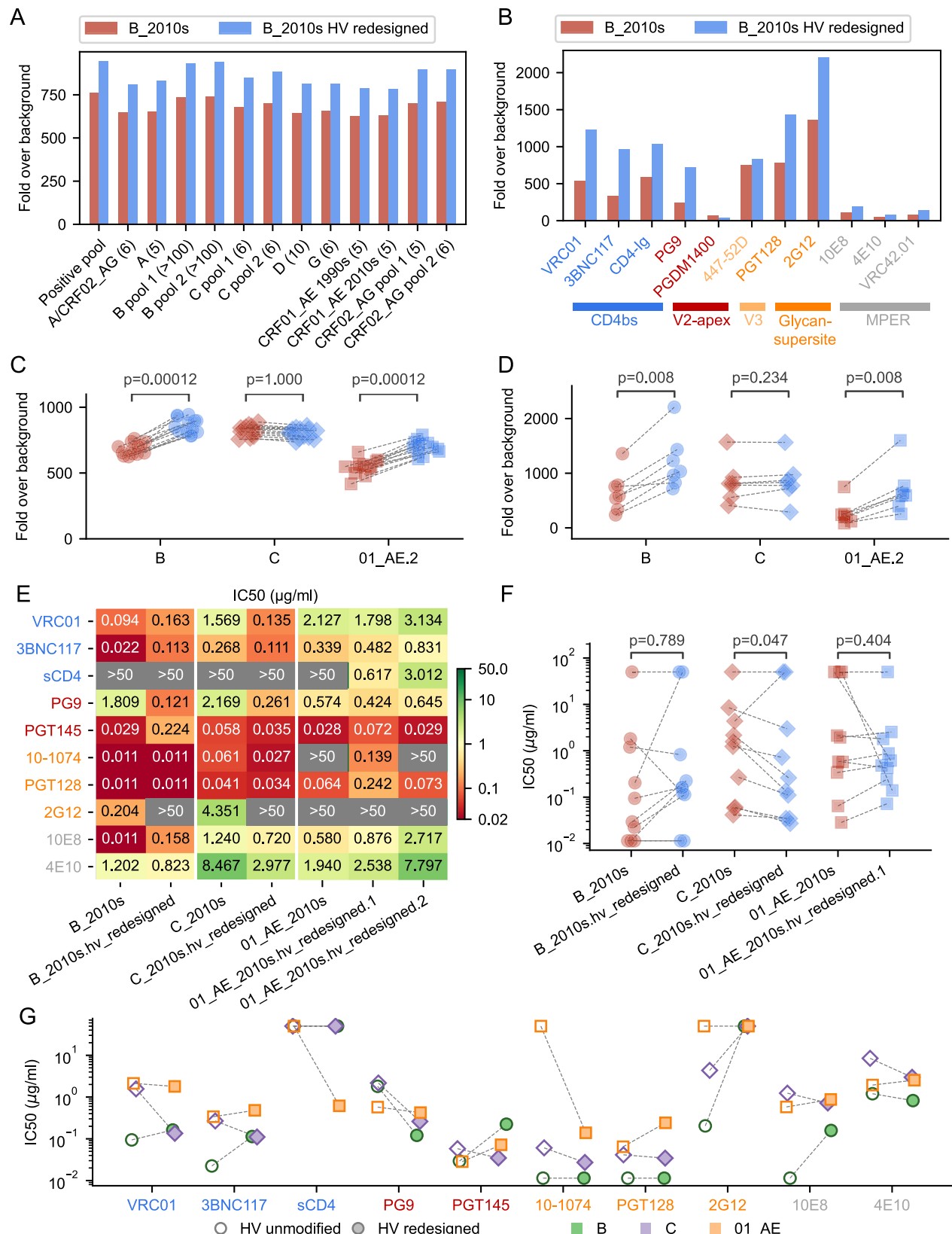

produced two decades ago. Consensus sequences were chosen because they represent the theoretically most fit virus, and, as such, they are more likely than a natural Env sequence to present favorable features for structural integrity as well as ideal bnAb epitopes[33–36]. We showed that our updated contemporary consensus sequences with or without the redesigned HV loops were functional as demonstrated by infectivity assays with pseudoviruses and EM analyses of the corresponding stabilized trimers. In addition, the consensus sequences with redesigned HV loops promoted antibody binding when compared to the unmodified updated consensus.

**Fig. 7 | Improved antibody binding to subtype B and CRF01_AE consensus Env with redesigned hypervariable (HV) loops.** Consensus Env sequences were expressed as gp140 glycoproteins. Comparison of the binding of **A** plasma pools from 13 cohorts of PLWH and **B** monoclonal antibodies to consensus B Env with unmodified (in red) and redesigned (in blue) HV loops. The number of plasma samples in each pool is shown in parenthesis in panel **A**. Binding of plasma samples ($n = 13$) **C** and monoclonal antibodies ($n = 7$, with PGDM1400 and MPER antibodies excluded) (**D**) to the subtype B, C and CRF01_AE consensus Env with unmodified HV loops (red) and redesigned HV loops (blue) is compared using paired, non-parametric one-tailed Wilcoxon signed-rank tests. MPER antibodies and PGDM1400 were not included in the statistical tests as their epitopes are not presented on the gp140 glycoproteins. **E** Neutralization sensitivity of the consensus Env with or without redesigned HV loops for subtypes B, C, and CRF01_AE measured for ten bnAbs that targeted the CD4 binding site (VRC01 and 3BNC117), V2-apex (PG9 and PGT145), glycan supersite (10-1074 and PGT128), MPER (10E8 and 4E10) and mannose-dependent (2G12) epitopes as well as the sCD4. The neutralization sensitivity is reported as IC50 values (μg/ml) with the most sensitive viruses colored in red and the most resistant in gray. Neutralization sensitivity (IC50 values) are shown for the unmodified and redesigned HV loops and displayed separately for each Env clade ($n = 10$) (**F**) or each bnAb (**G**). The increase in sensitivity with the redesigned HV loops is tested for each clade with one-tailed exact Wilcoxon-Pratt signed-rank tests. Only the 01_AE_2010s.hv_redesigned.1 is shown in panels **F** and **G** as it has a shorter V1HV than 01_AE_2010s.hv_redesigned.2. Source data for all panels are provided in the Source Data files.

Our analysis of a large set of sequences showed that Env with long HV loops were more resistant to bnAbs targeting cognate epitopes than Env with shorter loops. This is consistent with previous findings showing that the combined length of V1HV and V2HV was correlated with virus sensitivity to glycan-supersite or V3-glycan bnAbs, and that the length of V5 was related to virus sensitivity to CD4bs antibodies[52]. Interestingly, our structural modeling uncovered a novel mechanism for CRF01_AE's remarkable resistance to glycan-supersite antibodies such as 10-1074. It was previously proposed that 10-1074 resistance was mediated by a shift of the glycan at site 332 to site 334[52]. Here, we showed that our redesigned Env CRF01_AE consensus with the shortest V1HV (without the 'IGNITD' motif) was sensitive to 10-1074 despite having a glycan shifted from site 332 to 334 (Fig. 7E, F). The other (longer) redesigned CRF01_AE V1HV with the 'IGNITD' motif (also with no PNGS at site 332) was resistant to 10-1074, indicating that the long V1HV was associated with 10-1074 resistance. In our model of CRF01_AE V1HV N-anchor sites, the I147 in the 'IGNITD' motif was predicted to occupy the same location as the CDRH3 of 10-1074 (Fig. 6E and S5), a pocket composed of sites 328, 330, 415 and 417. In contrast, the pocket was not occupied by the modeled consensus B or C Env with unmodified V1HV. Furthermore, PGT128, a V3-glycan antibody whose epitope does not occupy this pocket[13] and is thus less occluded by the CRF01 AE V1HV, retains the capacity to neutralize half of the CRF01 AE viruses[52]. Our neutralization results confirmed that all the Env consensus sequences were sensitive to PGT128, while only the CRF01_AE with the shortest V1HV was sensitive to 10-1074 (the two CRF01_AE Envs that retained the 'IGNITD' motif remained resistant to 10-1074) (Fig. 7E, F). It is possible that other viruses with long V1HV use a similar mechanism to obstruct the binding of glycan-supersite bnAbs and divert antibody responses to these loops, highlighting that reducing the length of HV loops could play a key role in minimizing interference with bnAbs targeting adjacent epitopes. Importantly, the consensus sequences we used had HV loops of medium length due to how they are designed. Hence, redesigning the HV loop would likely have a more pronounced benefit in the context of a natural HIV-1 sequence.

Besides reducing the HV loop length, we focused on reducing the immunogenicity of the HV loop by using shorter spacers composed of fewer immunogenic amino acids. The redesigned spacers were composed mainly of glycine and serine residues except for the PNGS site. This is unlike natural HV loops, which consist of various AAs. Our goal was to minimize strain-specific responses. Since glycine and serine lack antibody-recognizable side chains, the redesigned loops are less likely to act as decoys attracting strain-specific antibody responses. A recent study found that long V1V2s in CRF01_AE Env were associated with the more rapid development of autologous neutralization in participants from the RV217 cohort[56]; these results highlight the potential role of long HV loops in the development of strain-specific responses – these strain-specific responses could be considered undesirable as autologous neutralization was not associated with the development of neutralization breadth in this cohort.

Importantly, we noted that HV loop length, particularly for V1, has not been an important consideration for vaccine inserts used in vaccine efficacy trials or under consideration for future trials (Table 1). Most of the sequences used in previous vaccines had HV loops that were longer than the median HV loop length in the population, suggesting that candidate inserts for future vaccine trials could be optimized to minimize the HV loop lengths to improve immunogenicity profiles. We want to emphasize that the HV loop redesign can and should be combined with other strategies. First, while bnAb epitopes near HV-loops were more accessible after our redesign, their accessibility may still be outcompeted within the immune system, with preferential recognition at strain-specific glycan holes or of the base of the trimer, which is common when a soluble, stabilized prefusion Env is administered[57–61]. Thus, fixing strain-specific glycan holes, and shielding the Env trimer base (e.g. by mounting it on a liposome[62] or co-delivering it with Gag in an mRNA vaccine[46]) could also help presenting bnAb epitopes. Second, the epitope near the redesigned HV loops may also need to be optimized, though what constitutes an optimal epitope for the elicitation of bnAbs is debatable. We previously showed that bnAb epitopes were actually as diverse as non-bnAb epitopes among HIV-1 sequences; however, what distinguished the broadest bnAbs from the other nAbs targeting similar epitopes was their ability to focus on key epitope sites that were more conserved[6]. As a result, if rare mutations are found in key epitope sites, it would make sense to revert the mutated residues to the consensus AA at that site. Third, many unshielded epitopes are exposed on the Env monomer and post-fusion Env. Thus, for the HV loop modification to be effective, a stabilized prefusion Env trimer is required as the improved bnAb accessibility may be insignificant in the monomer/post-fusion context. This can be achieved with the repair and stabilization procedure which systematically fixes rare mutations and stabilizes HIV-1 Env in the prefusion conformation[48,49].

There are some limitations to our computational approach to assess antibody accessibility. First, the glycan shield is absent while we calculate the depth of the epitope and the clashes between Env and bnAbs. That might lead to an overestimation of the antibody accessibility both before and after the HV-loop redesign, as the occlusion from glycans is omitted. Second, we modeled one subunit in the prefusion Env trimer by AlphaFold2 and modeled loop conformations based on them. The impact from neighboring subunits is absent in our evaluation. Third, when calculating the depth, a sphere with a radius of 30 Å was chosen as the antibody probe to simplify calculations, which is not necessarily a good approximation for Fv or Fab domain. As a result, our computational assessment of antibody binding should only be considered qualitatively correct in terms of capturing the antibody accessibility difference. Moreover, the neutralization sensitivity assays that compared how the different consensus proteins were neutralized by ten bnAbs showed that the Env with redesigned HV loops did not show increased sensitivity to all bnAbs across all subtypes. There were nonetheless striking effects regarding the increased sensitivity to PG9 across clades or the change from resistance to sensitivity to 10-1074 for

CRF01_AE Env. These results illustrate that Env sensitivity to neutralization is not an all-or-none phenomenon and warrants further studies. Although a significant increase in sensitivity was not evidenced across all bnAbs and all clades, it remains possible that our redesigned Env when used as immunogens could promote the immunofocusing of bnAb responses to conserved epitopes due to the elimination of potentially interfering residues. While the EM characterization did not reveal any aberration for our redesigned consensus Env with four HV loops shortened simultaneously, it will be important to test the effect of each HV loop redesign independently and to generate higher-resolution empirical structural information.

In this work, we redesign new Env consensus sequences that reflect currently circulating sequences for subtypes B and C and CRF01_AE with modified HV loops and use in vitro assays to test infectivity, neutralization sensitivity, and structural conformation to show that these computationally-derived sequences resulted in functional Env. We demonstrate that long HV loops interfered with bnAbs targeting adjacent epitopes and that our modifications alleviate the interference of HV loops on bnAb epitopes, especially for the glycan-supersite bnAbs. Since half of all neutralizing antibodies target epitopes around the four optimized HV loops[22,23], we propose that this design strategy could yield more effective antigens than those currently used as reagents or vaccine candidates. This optimization strategy can be integrated with other vaccine design techniques, such as the 'repair and stabilize' of Env in the prefusion conformation[48,49], shielding the base of Env[46], or the presentation of an optimum combination of Envs[63], to create the next-generation of HIV-1 vaccines for eliciting broadly protective antibodies.

## Methods

### Dataset
Env AA sequences of subtypes B, C, and CRF01_AE were retrieved from the Los Alamos National Laboratory HIV-1 sequence database on 2022-11-07. Sequences were retained if they had a complete open reading frame, information on the year and location of collection, and were not hypermutated or problematic (as defined by LANL). Non-independent sequences were removed so that the final dataset included one sequence per person. The IC50 of PGT145, VRC26.25, 10-1074, PGT135, VRC01, and 3BNC117 against viruses that had the corresponding Env sequences available were downloaded from the CATNAP database as of 2022-10-04[47]. In total, there were 549, 423, 782, 310, 1032, and 818 IC50s and associated Env sequences for PGT145, VRC26.25, 10-1074, PGT135, VRC01 and 3BNC117, respectively.

### HV loop analysis
The HXB2 genome annotation from LANL HIV Database (https://www.hiv.lanl.gov/content/sequence/HIV/MAP/annotation.html) was followed to define the start and end of HV loops, which were 132-152, 185-190, 396-410 and 460-467, for V1HV, V2HV, V4HV and V5HV, respectively. As an exception, V4HV was extended to include 395 and 411-412 anchor sites in the HV loop redesign. Structurally, the distance between the start and end of spacers was calculated as the distance between CA atoms before the N-terminal and after the C-terminal of the spacer loop. The HXB2 sequence was added and aligned to define the positions. MAFFT[64] was used to align sequences. For the logo plot of HVs, HV loops were split from the middle: the first half was left-aligned to the start position while the second half was right-aligned to the end position.

### Structure prediction and analysis
We used local ColabFold[65] which integrated AlphaFold-Multimer[66] with other tools and databases[67–72] to model unmodified and redesigned Env structures. During the prediction, one of the three subunits in the prefusion Env trimer is predicted by feeding one gp120 (31-511) and one gp41 (512-665) as input. The top model from the five predictions

was adopted for later analysis. Loop modeling was done with the KIC loop modeling protocol[50] via PyRosetta[51]. The four HV loops were modeled sequentially in random orders in individual runs.

The non-HV surrounding surface sites of HV loops were defined as Env sites that were not part of the HV loop which have relatively accessible surface area (RSA) greater or equal to 0.30 and have any atoms within 10 Å of any HV loop atoms in the predicted unmodified consensus B Env structure. The accessible surface area (ASA)[73] was approximated by the Shrake-Rupley method[74]. In brief, evenly distributed points ($n = 256$) were placed on spheres that were centered at atoms in the protein. The radius of the spheres was the sum of the atoms' and the solvent probe's radius (1.5 Å). The accessible surface was defined after filtering out points on a sphere that were within neighboring spheres. Then, the RSA of a site was calculated as the ratio of ASA over the estimated maximum ASA[75]. The same approach was also used to estimate depth, only the probe radius was set to 30 Å, and the depth equals the minimal distance between a protein atom and the accessible surface minus the probe radius. Env structure 5FYJ[76] was used for the ASA estimation. It was also used in the illustration of the HV loop and nearby bnAb epitopes in Fig. 1.

The clashes between antibody and Env were determined as the number of antibody atoms within 2.5 Å of Env after superimposing the modeled Env on the Env in the solved Env-antibody complex. Bio.PDB[77] in Biopython[78] was used to superimpose structures. The Env-antibody complexes used, listed as PDB code, were 5V8L[17], 6VTT[18], 6UDJ[79], 4JM2[12], 6V8X[80] and 7PC2[81] for PGT145, VRC26.25, 10-1074, PGT135, VRC01, and 3BNC117, respectively. PyMol[82] was used for the structure visualization and figure generation. Hydrogen atoms, when present, were removed from the structure, in all analyses.

### Protein production and antibody binding
Uncleaved gp140 proteins were designed with R to S mutations in the gp120/gp41 furin cleavage site and incorporated the native leader peptide and full MPER sequence, followed by a short GGGS linker sequence and a C-terminal AviTag. Sequences, codon-optimized for expression in human cells, were synthesized (Genscript) and cloned into a custom pcDNA3.4 (ThermoFisher) expression vector (plasmids are available with a Material Transfer Agreement). Proteins were expressed by transient transfections in Expi293F cells (ThermoFisher) according to the manufacturer's instructions. Gp140 proteins were purified from clarified cell culture supernatants 4 days post-transfection using Galanthus nivalis lectin (GNL) affinity chromatography followed by a Q-sepharose (GE Healthcare) polishing step to remove host cell protein contaminants. All gp140s were purified to 90% purity or higher, as assessed by SDS-PAGE and Coomassie staining in reducing and non-reducing conditions. Antibodies VRC01[20], 3BNC117[83], CD4-Ig[84], PG9[19], PGDM1400[15], PGT128[13], 2G12[85], 10E8[86], 4E10[14,87] were produced in-house from public sequences as IgG1 by transfection in Expi293F cells and purified by protein A affinity chromatography; antibodies 447-52D[88] and VRC42.01[89] were obtained from Polymun (Cat. No. AB014) and from Nicole Doria-Rose (VRC, NIH), respectively. For antibody binding and characterization, gp140 proteins were coupled to uniquely coded carboxylated magnetic microspheres (Luminex Corp., Austin TX) per manufacturer's protocol and as previously described and pooled into an antigen cocktail. Antibody binding and characterization were performed as previously described[53,90,91] with minor modifications. The antigen cocktail was incubated with an array of heat-inactivated human plasma pool samples from various clade-specific cohorts of PLWH, or with monoclonal antibodies targeting various regions of HIV-1 Env. Samples were incubated for 2 hours at room temperature, washed then incubated for 1 hour with phycoerythrin labeled anti-human IgG Fc (Southern Biotech, Birmingham AL) detection antibody. After a final wash to remove unbound detection reagent, microspheres were resuspended in 40 μL sheath fluid (Luminex Corp) and data was collected on a Bio-Plex®3D

Suspension Array system (Bio-Rad, Hercules CA) running xPONENT® v.4.2 (Luminex Corp). A multi-clade control plasma pool comprised of samples from HIV-1 Group O, subtypes B, C, and CRF01_AE, CRF02_AG, and CRF03_AB (SeraCare, Milford MA) was used as a positive control ("Positive Pool") and a pool of plasmas from people without HIV-1 (PWOH) served as a negative control. The figures correspond to monoclonal antibodies at a concentration of 10 ug/mL and pooled plasma at 1:100 dilution. Responses are reported as fold-over PWOH for plasma samples and fold-over the ZIKA virus-specific monoclonal antibody MZ4[92].

### HIV-1 Env expression and antibody binding

293 T cells were transfected with 1 ug of plasmid DNA encoding an HIV-1 env gene (PolyJet™ In Vitro DNA Transfection Reagent (SignaGen Laboratories, Cat. No. SL100688) in 6-well plates and harvested at 48 h. Cells were split into separate staining reactions for monoclonal antibodies 3BNC117 (Cat. No. ARP-12474), 10-1074 (Cat. No. ARP-12477), 10E8 (Cat. No. ARP-12294), 447-52D (Cat. No. ARP-4030, all from BEI resources), b12 (Polymun, Cat. No. AB011), PGDM1400[15] and PGT151[93] (both kindly provided by Dr. Devon Sok and International AIDS Vaccine Initiative). 2G12 was produced in Expi293F cells from publicly available heavy and light chain sequences[94] and purified using Protein A affinity chromatography. Cells were washed and stained with anti-human IgG-PE at 1:250 dilution (SouthernBiotech, Cat. No. 2040-09) for 15 min and fixed in 0.5% paraformaldehyde. Fluorescence was measured on a FACSymphony A5 SE Cell Analyzer and analyzed in FlowJo v.9.9.6.

### Recombinant HIV-1 Env stabilized trimer production and purification

Stabilized, soluble trimers in a prefusion state were engineered by introducing previously reported[48,49] and additional modifications with ten stabilization mutations (M302, V304, L320, N535, P556, E588, V589, F651, I655, and V658) and three SOSIP mutations (C501, C605 and P559). Sites 508-511 were replaced with six arginines for more efficient furin cleavage. All Envs were terminated at site 664. Trimers were expressed in Expi 293 cells (Thermo Fisher Scientific) by transient transfection with plasmid DNA complexed with PEI. 800 µg of the HIV-1 Env expressing plasmid and 200 µg of furin expressing plasmid were used for each 1 L of cells. Cell-free culture supernatant was 0.2 µm filtered and purified by GNA Lectin (VectorLabs) before size exclusion chromatography on a Superdex 200 16/600 column (GE Healthcare Cat. No. 29323952) in PBS. Any shoulders to the roughly monodisperse Env trimer peak were removed, and protein was pooled and concentrated before adding glycerol to a final concentration of 5%. Trimeric HIV-1 Env protein was sterile-filtered, snap-frozen, and stored at -80 °C before further use.

### Negative-stain Electron Microscopy

Purified protein was thawed and diluted to 10 µg/ml with 20 mM HEPES, pH 7.5; 150 mM NaCl immediately prior to application on glow-discharged carbon-coated grids (Electron Microscopy Sciences #CF300H-Cu-50) and stained with 2% phosphotungstic acid, pH 7.4. Samples were imaged on a JEOL CryoARM 200 using SerialEM 4.1.0beta with a nominal magnification of 60kX and a pixel size of 0.873 Å/pixel on a Ametek Gatan K3 in counting mode. A 100-frame movie with a total dose of 225 electrons/Å$^2$ was motion corrected and CTF was estimated using patch CTF in CryoSPARC v4[95]. All further processing was done in Cryosparc v4. Particle picking was conducted with a circular Gaussian blob of 100-150 Å for all samples. Reference-free 2D classification was used to separate false picks from putative particles. A second round of 2D with a maximum alignment resolution of 15 Å and a maximum resolution of 10 Å on putative particles was used to separate monomer and trimer classes and estimate the relative percentages of each species. For the gp140 samples, no averageable trimer was observed.

For stabilized trimer samples, A C1 ab initio model was created for particles that classified as trimers in 2D for each of the protein samples imaged. This was used as an initial model for a C1 3D homogenous refinement with an initial low pass filter of 60 Å, GS-FSC split of 40 Å, and a maximum alignment resolution of 20 Å. The resulting consensus structure was 3D classified without alignment with 10 classes and a target resolution of 15 Å. The particles from the most open trimer class and most closed trimer class were selected and refined with 3D homogenous refinement using the volume from their classification with the same parameters as noted above in both C1 and C3. C1 and C3 closed and open models were then used as initial models for a two-class heterogenous refinement with an initial resolution of 30 Å and a maximum resolution of 15 Å. The relative percentages and maps shown were estimated using this C1 heterogeneous refinement, but C3 refinements converged on the same approximate proportions.

This procedure was effective for all samples except for the subtype C consensus stabilized trimer without HV loop redesign. In this case, the classification without alignment did not produce a particle subset that was closed enough to separate closed from open trimer using heterogenous refinement. In this case, an initial model of closed trimer from the subtype C stabilized trimer with HV loop redesign sample and an open trimer without HV loop redesign was used for the final C1 heterogenous refinement. Data was visualized and figures were generated using ChimeraX[96] and EMAN2[97].

### HIV-1 pseudovirus (PSV) production, titration and neutralization

HIV-1 PSVs were produced by co-transfection of HEK293T/17 cells (ATCC, Manassas, VA, USA) with a pSG3ΔEnv DNA plasmid encoding the HIV-1 backbone (24 µg DNA) and a plasmid encoding the HIV-1 env gene (8 µg DNA). Plasmids were combined with X-tremeGENE 9 DNA transfection reagent (Sigma-Aldrich, St. Louis, MO, USA) at a 1:3 ratio in serum-free DMEM, per manufacturer recommendations. The transfection mix was added to a T75 flask containing plated HEK293T/17 cells in a culture medium. The flask was incubated for 6 hours at 37 °C. Then the media was replaced. PSVs were harvested 72 hours after transfection, filtered through a 0.45 µm PVDF membrane (MilliporeSigma, Burlington, MA, USA), and stored in 3 ml aliquots at -80 °C.

PSV stocks were titrated on TZM-bl cells (NIH AIDS Research and Reference Reagent Program, Cat. no. 8129) to determine an acceptable dilution for neutralization assays. Virus stocks were serially diluted then added to 96-well plates (Thermo Fisher Scientific, Waltham, MA, USA) in a final 25 µL/well volume. An additional 25 µL of supplemented DMEM was added to each well. TZM-bl cells were added to plates in 50 µL of supplemented DMEM with 40 µg/mL of DEAE-Dextran (Sigma, St. Louis, MO, USA). Virus and cells were incubated together for 48 hours at 37 °C. Infectivity was measured by luminescence after adding 100 µL of the Bright-Glo Luciferase Assay System (Promega, Madison, WI, USA) to each well; luminescence was quantified using the EnVision luminometer (Perkin Elmer, Waltham, MA, USA).

Neutralizing antibody titers were determined using TZM-bl cells in a manual 96-well assay. BnAbs were obtained from the HIV reagent Program (VRC01 (Cat. No. ARP-12033, Lot 180231); 3BNC117 (Cat. No. ARP-12474, Lot 140211); sCD4 (Cat. No. ARP-4615, Lot 180011); PGT145 (Cat. No. ARP-12703, Lot 190527); 10-1074 (Cat. No. ARP-12477, Lot 190442); PGT128 (Cat. No. ARP-13352, Lot 190522); 2G12 (Cat. No. ARP-1476, Lot 180138); 10e8 (Cat. No. ARP-12294, Lot 170395)0 or from PolyMun (PG9 (Cat. No. AB015, Lot PG9_021014A); 4e10 (Cat. No. AB004, Lot T531002-A)). BnAbs were prepared at a starting concentration of 25 µg/mL in supplemented DMEM. PSVs were diluted to a concentration to yield 100,000 relative light units (RLUs) and at least 3 times above the cell-only controls, as determined by the PSV titration assay. Neutralization reagents were serially diluted. Titered bNAbs were transferred to 96-well culture plates and incubated with an equal volume of PSV for 1 hour at 37 °C. A non-specific virus control (MuLV)

was used to ensure neutralization by the bNAbs was HIV-specific. TZMbl cells ($1 \times 10^4$ cell/well) mixed with 40 µg/mL DEAE-dextran (Sigma, St. Louis, MO, USA) were added to each well and incubated for an additional 48 hours. RLUs were detected with the EnVision luminometer (Perkin Elmer, Waltham, MA, USA) using the Bright-Glo Luciferase Assay System (Promega Corporation, Madison, WI, USA). Percent neutralization (percentage reduction of relative light units in the presence of bNAb) was calculated for each bNAb concentration. Neutralization dose-response curves were fitted by nonlinear regression using the LabKey Server, and the final titer is reported as the concentration of bNAb (IC50/IC80) necessary to achieve 50% or 80% neutralization.

### Quantification and statistical analysis

Statistical analyses were performed using SciPy[98] and coin[99]. The non-paired, non-parametric Mann-Whitney U tests were used to compare the IC50 of short/long nearby or distant HV loops in Fig. 2, S2 and S3. The paired, non-parametric Wilcoxon signed-rank tests were used to compare the depth difference before and after HV loop redesign (Fig. 6) and the binding responses before and after the HV loop redesign with pooled plasma samples or bnAbs (Fig. 7). The one-tailed exact Wilcoxon-Pratt signed-rank test is carried out using the coin package[99]. Differences corresponding to p-values of less than 0.05 were considered as significant.

## Data availability

The source data for all figures and tables in the manuscript and in the Supplementary Information are provided with this paper. Source data are provided with this paper.

## Code availability

The code used for the analysis is available at https://www.hivresearch.org/node/951.

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

## Acknowledgements

We thank Johannes PM Langedijk and Lucy Rutten for helpful discussion, Indie Showell-DeLeon and Diana Wasson for technical assistance. We thank Phuc Pham for checking the code. This work was supported by a cooperative agreement between The Henry M. Jackson Foundation for the Advancement of Military Medicine, Inc., and the U.S. Department of the Army [WW81XWH-18-2-0040] and by funds from the NIH (1R01AI176520-01) to SV.

## Author contributions

H.B. and M.Ro. conceived the study, analyzed the results and wrote the manuscript. H.B. and P.V.T. designed and performed analyses. E.L. and Y.L. curated sequence alignments and designed consensus sequences. P.V.T. and M.G.J. characterized redesigned Env in vitro. M.Z., M.M., C.E.P., T.T., P.A.R., A.H., V.D., B.S., L.M.-R., A.S., E.K., M.Ra., G.S., J.F., A.S., L.W., V.P., S.J.K, D.L.B., S.T. produced Env proteins and mapped antibody responses. J.A.A. and S.V. provided support and funding.

## Competing interests

The views expressed are those of the authors and should not be construed to represent the positions of the U.S. Army, the Department of Defense, or the Department of Health and Human Services. A patent application on the invention disclosed in this publication is filed. Hongjun Bai, Eric Lewitus and Morgane Rolland are the co-inventors. Other authors declare no competing interests.
