## [Peer Review File · Nature Communications]

Contemporary HIV-1 consensus Env with redesigned hypervariable loops promote antibody bindingREVIEWER COMMENTS

Reviewer #1 (Remarks to the Author):

This paper describes the redesign of clade B, C and AE consensus Envs to improve recognition by V2, V3-glycan and CD4bs bNAbs by eliminating clashes. To assist in designs, this group used sophisticated methods including AlphaFold2 to predict structures and design spacers often shorter and enriched with GS residues. Also, they used computational models of depth index to determine mAb-antigen compatibility. They also used updated consensus sequences. In the end, they identified clones with redesigned V1, V2, V4 and V5 loops, which, in some cases (V1 especially) are shorter and in other cases are GS rich, to dampen immunogenicity and presumably grant flexibility. Overall, the paper is interesting, well done and well-written. It will be interesting to know if this strategy can in future help in our collective quest to induce bnAbs.

The authors mention that the use of consensus redesign described here needs to be combined with other strategies such as resurfacing target epitopes, removing glycan clashes, filling glycan holes and repair and stabilize. It occurs that the consensus candidates reported could be potentially used as priming immunogens. Boosts might then have variations in variable parts of target epitopes to boost cross-reactivity by forcing increased focus on remaining conserved residues. Boosts might also have longer loops restored, so that nascent NAb can learn to navigate the structures it will need to encounter in the "wild" to broadly neutralize.

Related to context, it would be interesting to know if the consensus Env sequences and their redesigned versions are infectious as pseudoviruses. If so, it would be very interesting to know how the changes affect neutralization sensitivity. The significance of such data would be greater than that with gp140s examined in Fig. 6, as recognition would be anchored in functional Env. It might also be of interest to assess each modified loop alone in pseudovirus as well as in concert with the others. Besides from the unpredictable effects of combining the redesigns in trimers, if they were used alone they might be able to immunofocus to a particular epitope cluster, while leaving clashes at other epitope clusters to reduce competition and thereby increase focus on a desired target(s), that may be more practical than going for everything at once.

Did the authors look at the effects of varying V1 length for V2 bNAbs? Q23 is among the most sensitive V2 strain and also has an unusually short V1, which might be no coincidence.

Can the authors speculate as to why the redesign didn't seem to improve antibody binding to clade C in Fig. 6?

Minor points:

Abstract: V2H in abstract should be V2HV and line 29 could say "loops" after V5HV. Sentence could be "We reduced the lengths of V1, V2 and V5 HVLs while..."

Lines 30 and 31. Consecutive sentences start with "Redesigned". It is not clear from reading this (at this stage in the manuscript) where the GS residues would come in when loops are shortened, as residues may be removed, not replaced? This just needs adjusting for clarity. Perhaps this applies largely to the V4 modifications. The use of GS residues might also increase flexibility?

Abstract line 31 passive voice could be made active so sentence could read: "Pooled HIV-1 positive plasmas and representative bnAbs showed improved binding to modified subtype B and CRF01_AE Envs". Does the abstract also need to mention what happened to the subtype C in this case (binding unchanged?).

Line 77: "propose" suggests it was not yet done! Suggest that this sentence is adjusted to something like "To overcome..., we systematically redesigned HV loops..."

Results line 96. 4 consecutive sentences start with "We..". This could be adjusted for better readability?

Line 151. P value for 10-1074 for V5HV is 0.838 based on fig 2C not 0.207 (that is VRC26). Also check the p values called out for Fig S2 (line 157). PGT145 should be 0.293 not 0.101 and VRC26 not mentioned in text could be called out as 0.079 (nearly significant).

Line 182, callout Fig S3 to S10

Line 258, callout should be to Fig. S14 not S5?

Line 293. Starting a sentence with "And..." is a bit unusual. Could be replaced by "Furthermore.." or something like that.

Fig. 6 binding was done by what appears to be a multiplex flow cytometry method, but this isn't mentioned in either the results narrative or the figure legend, but probably should be.

Fig 6 labels of epitope are tough to follow with diagonal mAb labels and epitope clusters. Given the result of 10-1074 in Fig 5E, how did that translate to binding differences (10-1074 is not in fig 6 or S14?).

Reviewer #2 (Remarks to the Author):

Current HIV-1 vaccine designs using the prefusion closed form mainly focused on stabilization of the prefusion conformation, targeting specific germline precursors or glycosylation forms. The effects of variable loops which form or surrounding major sites of vulnerability are seldomly considered. The manuscript "Contemporary HIV-1 consensus Env with redesigned hypervariable loops promote antibody binding" by Bai and colleagues utilized from HIV-1 database, analyzed the length distribution of variable loops V1, V2, V4 and V5 in HIV-1 gp120 and the effects on plasma antibody and monoclonal antibody binding and neutralization, they employed AlphaFold modeling to model gp120 monomers with different length of variable loops. Based on these data, the authors designed HIV-1 Env with consensus sequences and altered hyper variable loops and showed that these designed proteins increased magnitude of binding responses to pooled plasma samples and representative bnAbs.

Overall, this is a well-designed study and the results added new information needed for the design of an effective HIV-1 vaccine. The study could be improved if the authors could perform some of the additional characterization for the developed immunogens:

1. What are the conformation of the designed Env proteins? A simple negative EM with 2D and 3D classification might confirm the conformational states.
2. How is the glycan shield looking after modifying the HVs? Could the authors perform a glycan analysis on the expressed trimer?
3. Are these Env expressed without commonly used SOSIP stabilization?
4. The authors used R to S mutations to disable furin cleavage. How did the authors make sure that the C-term of gp120 and N-term of gp41 are in correct locations? We know from the prefusion closed form that the end of gp120 and the N-term of fusion peptide are located apart.

Reviewer #3 (Remarks to the Author):

Bai et al. report a new modeling-based strategy for designing Envs with improved bnAb binding profiles, focusing on shortening the hypervariable loops V1, V2, V4, and V5. The authors rely on AlphaFold2 for building models of gp120s for designed Envs with shortened hypervariable loops from 3 different clades of HIV and test a subset of the designs with experimental binding assays. The results presented could be informative for guiding the design of hypervariable loops in immunogens, however, it is hard to come to definitive conclusions about the data as it is currently presented. My comments focus on clarifying statistical results:

1. In the boxplots in Figure 1, the meaning of the whiskers is not defined. This needs to be clarified in the figure legends so that the figures can be accurately interpreted by the reader.
2. Figure 2 and related supplementary figure 1 does not define the meaning of the whiskers in the boxplots. Furthermore, supplementary figure 1 does not describe the boxes, centrality, or distributions shown in the boxplots.
3. For supplementary figures related to figures 3 and 4, some are missing either the N-anchor or C-anchor and it is unclear why those are not needed for some of the designs. Explaining why N/C-anchor was/were not used would help to clarify the design strategy.
4. It's unclear where the replicate values in Figure 5B-D are coming from.
5. Figure 2D V1HV long is missing the whiskers and there is no explanation. Whiskers are also missing from supplementary figure 1A PGT145-other V2HC long, 1B VRC26.25-B V2HV short and long and V5HV long, and 1D PGT135-C V1HV short and long. Add an explanation of the whiskers and why they are missing from some plots.
6. In figures 2B, 2C, 4A, and supplementary figures 1A, 1B, 1C, 1D, 1E, and 1F some of the whiskers do not include the full spread of the data, which can only be if they considered those points outliers but outlier detection methods were not described in the methods or figure legends.
7. In Figure 2 and supplementary Figures 1 and 2 all IC50 values over 100 ug/mL are grouped. However, this makes interpreting the boxplots difficult. The position of the median does not make sense in some cases because $\geq 50\%$ of the points occur at lower IC50 values than where the median is marked. This trend occurs more often in the long-loop box plots than in the short-loop plots. Plot the full distribution, including on IC50 ≥ 100 ug/mL, so that the reader can more easily interpret the statistics described by the boxplots.
8. There are no methods for outlier detection/removal listed but it seems like outliers were excluded. In some of the boxplots, the whiskers do not include the full range of data, suggesting that these points were excluded as outliers. This is only true for points with low IC50 values and this occurs more frequently in Envs with long VH loops. Potential outliers with high IC50 values do not seem to have been considered because all values over 100ug/mL are grouped. If only low IC50 values were analyzed for outliers, without checking for high outliers, making the outlier detection one-sided, this skews the entire analysis towards higher IC50 values. Since this occurs more frequently in Envs with long VH loops I am concerned that the analysis skews the results towards supporting the hypothesis that shorter loops improve bnAb binding. Providing a description of the outlier detection and removal process and plotting the full distribution of the points ≥ 100 ug/mL would strengthen the arguments and provide more robust support for the claims that shorter HV loops improve bnAb binding.
9. Lines 175-176 state that D133 interacts with K154 but is interacting with K155 in Figure 3.
10. The section describing the HV loop redesign is vague and lacks critical information. Some of the interactions discussed do not make interactions in the structures presented in the figures. Was more than one conformation used to calculate the structures? Other potential interactions shown in the figures are not discussed in the text (ex: K151 L134/D133). Discussion of important

interactions considered in the redesign of other loops is also missing. Furthermore, it is unclear how the length of the linker in the design was chosen. The linker is different for every designed construct but there is no information about how the specific number of amino acids was chosen. The missing details obfuscate how the design strategy was carried out. Adding a more thorough discussion of the interactions and reasoning behind the linker design used in the HV design strategy would greatly strengthen the claims and importance of the findings.

11. AlphaFold2 still struggles to provide accurate structures for unstructured loop regions in proteins, especially for domains that can exhibit multiple conformations, yet this method was used to model the flexible and unstructured HV loops in Env. The structures used in this accessibility analysis may not be reliable. The fit scores from AlphaFold2 should be provided so that the reader can assess the quality/validity of the models.

12. A sphere probe was used to assess the antibody accessibility in redesigned HV loop models, and this was followed up with binding studies on a select few designs, however, there was no correlation shown between the modeling and experimental results. One analysis that could strengthen this paper would be to include a regression analysis between the antibody accessibility predicted by the modeling and the experimental antibody binding data. Showing a correlation between these two data sets would establish the predictive value of the modeling and design strategy.

Please see below for our responses in blue.

Reviewer #1 (Remarks to the Author):

This paper describes the redesign of clade B, C and AE consensus Envs to improve recognition by V2, V3-glycan and CD4bs bNAbs by eliminating clashes. To assist in designs, this group used sophisticated methods including Alphafold2 to predict structures and design spacers often shorter and enriched with GS residues. Also, they used computational models of depth index to determine mAb-antigen compatibility. They also used updated consensus sequences. In the end, they identified clones with redesigned V1, V2, V4 and V5 loops, which, in some cases (V1 especially) are shorter and in other cases are GS rich, to dampen immunogenicity and presumably grant flexibility. Overall, the paper is interesting, well done and well-written. It will be interesting to know if this strategy can in future help in our collective quest to induce bNAbs.

The authors mention that the use of consensus redesign described here needs to be combined with other strategies such as resurfacing target epitopes, removing glycan clashes, filling glycan holes and repair and stabilize. It occurs that the consensus candidates reported could be potentially used as priming immunogens. Boosts might then have variations in variable parts of target epitopes to boost cross-reactivity by forcing increased focus on remaining conserved residues. Boosts might also have longer loops restored, so that nascent NAbs can learn to navigate the structures it will need to encounter in the “wild” to broadly neutralize.

Related to context, it would be interesting to know if the consensus Env sequences and their redesigned versions are infectious as pseudoviruses. If so, it would be very interesting to know how the changes affect neutralization sensitivity. The significance of such data would be greater than that with gp140s examined in Fig. 6, as recognition would be anchored in functional Env. It might also be of interest to assess each modified loop alone in pseudovirus as well as in concert with the others. Besides from the unpredictable effects of combining the redesigns in trimers, if they were used alone they might be able to immunofocus to a particular epitope cluster, while leaving clashes at other epitope clusters to reduce competition and thereby increase focus on a desired target(s), that may be more practical than going for everything at once.

As suggested by the reviewer, we conducted additional experiments. We generated pseudoviruses with the Env consensus sequences with or without the hypervariable loops redesigned. The revised manuscripts include results from infectivity and neutralization assays as shown in Fig. 7. Moreover, we also designed stabilized trimers that correspond to the consensus Env and show electron microscopy results. These results showed that all the computationally derived Env were infectious as pseudoviruses and that the Env with redesigned shorter hypervariable loops had similar structural conformations as our original consensus Env. The neutralization sensitivity for ten bNAbs against pseudoviruses showed mixed and non-consistent results for the redesigned HV loops, as neutralization sensitivity was neither increased or decreased across all ten bNAbs. These results warrant future work to define the effect of each HV loop modification independently.

Did the authors look at the effects of varying V1 length for V2 bNAbs? Q23 is among the most sensitive V2 strain and also has an unusually short V1, which might be no coincidence. We thank the reviewer for this suggestion. We updated the analysis to include the three other HV loops when comparing the impact of a given loop length on IC50 measurements and modified the Fig. 2 accordingly. As suggested by the reviewer, V1HV length showed a significant effect on IC50 values for V2-apex antibodies; the impact of V1HV on V2-apex

antibodies was similar to that of V2HV suggesting that the interaction between V1HV and V2HV could affect the accessibility of V2-apex epitope.

We revised the manuscript as follows:

'To evaluate the effect of loop length on bnAb sensitivity, we compared short and long HV loops for the HV loop proximal to each bnAb (V2HV to V2-apex antibodies, V1HV to glycan supersite antibodies, and V5HV to CD4bs antibodies) and for the three other HV loops. The virus sequences with the longest adjacent HV loops had significantly higher half maximal inhibitory concentration (IC₅₀) than the sequences with the shortest HV loops (p-value \leq 0.015, one-tailed Mann–Whitney U test) for the six bnAbs (Fig. 2), suggesting that longer HV loops obstruct bnAb epitope availability. When HV loops distant from the bnAb epitope (V5HV to V2-apex and glycan supersite antibodies and V2HV to CD4bs antibodies) were compared, there was no significant difference in sensitivity for short or long HV loops for five of the six bnAbs tested (p-value \geq 0.207). Interestingly, in specific cases there might be some longer-range interactions between HV loops. Sequences with long/short V1HV had a similar impact on IC₅₀ values for V2-apex antibodies as V2HV, suggesting that interactions between V1HV and V2HV may also modulate the accessibility of V2-apex epitopes.'

Can the authors speculate as to why the redesign didn't seem to improve antibody binding to clade C in Fig. 6?

It is unclear to us what may cause this. We note, however, that the subtype C Env already showed high plasma pool binding before the modification of HV loops, while we note that the pseudovirus neutralization titers were improved. The following was added to the text:

'The subtype C consensus proteins with or without redesigned loops showed no difference in antibody binding (Fig. 6CD, Fig. S15), knowing that the consensus C Env already showed high plasma pool binding relative to the other subtype/CRF before the loops were redesigned.'

Interestingly, the neutralization sensitivity assays across the ten bnAbs showed that 'the subtype C Env with redesigned HV loops was more sensitive to the ten bnAbs than the subtype C Env with unmodified loops (p=0.047 for IC₅₀ values and p=0.016 for IC₈₀ values).'

Minor points:

(1) Abstract: V2H in abstract should be V2HV and line 29 could say "loops" after V5HV.

Sentence could be "We reduced the lengths of V1, V2 and V5 HVLs while..."

We thank the reviewer for this suggestion. The text was revised accordingly.

(2) Lines 30 and 31. Consecutive sentences start with "Redesigned". It is not clear from reading this (at this stage in the manuscript) where the GS residues would come in when loops are shortened, as residues may be removed, not replaced? This just needs adjusting for clarity. Perhaps this applies largely to the V4 modifications. The use of GS residues might also increase flexibility?

We modified the text to state that this applied to V1 and V4 HV. The use of 'GS' is indeed expected to make the HV loop flexible, which is beneficial for the structural compatibility. The revised sentence is as follows:

'Across all redesigned HV loops, the spacer consisted mainly of glycine and serine which lack the side chains that are preferentially targeted by antibodies and may also increase the flexibility of the HV loops.'

(3) Abstract line 31 passive voice could be made active so sentence could read: "Pooled HIV-1 positive plasmas and representative bnAbs showed improved binding to modified subtype B and CRF01_AE Envs". Does the abstract also need to mention what happened to the subtype C in this case (binding unchanged?).

We modified the abstract accordingly: ‘Pooled plasma samples from people living with HIV-1 and representative bnAbs showed improved binding to modified subtype B and CRF01_AE Env but not to subtype C Env.’

(4) Line 77: “propose” suggests it was not yet done! Suggest that this sentence is adjusted to something like “To overcome..., we systematically redesigned HV loops...”

We revised the text accordingly. Thanks for the suggestion.

(5) Results line 96. 4 consecutive sentences start with “We..”. This could be adjusted for better readability?

We revised the text and removed three ‘we’.

(6) Line 151. P value for 10-1074 for V5HV is 0.838 based on fig 2C not 0.207 (that is VRC26). Also check the p values called out for Fig S2 (line 157). PGT145 should be 0.293 not 0.101 and VRC26 not mentioned in text could be called out as 0.079 (nearly significant).

We thank the reviewer for pointing these errors. The description of Fig. S2 was updated:

‘Differences between short/long adjacent HV loops remained statistically significant for most bnAbs ($p < 0.045$, Fig. S2C-F) or borderline significant for VRC26.25 when the cutoff was set as 25% ($p = 0.079$, Fig. S2B), with PGT145 as an exception ($p \geq 0.101$ with 10-25% as the threshold (Fig. S2A).’

(7) Line 182, callout Fig S3 to S10

This was corrected.

(8) Line 258, callout should be to Fig. S14 not S5?

This was corrected.

(9) Line 293. Starting a sentence with “And...” is a bit unusual. Could be replaced by “Furthermore..” or something like that.

This was modified.

(10) Fig. 6 binding was done by what appears to be a multiplex flow cytometry method, but this isn’t mentioned in either the results narrative or the figure legend, but probably should be.

We used a bead-based multiplex immunoassay (‘Luminex’) that we previously set up (Mdluli et al., PloS Pathogens, 2020; Merbah et al., JIM, 2020). This was added in the Results.

(11) Fig 6 labels of epitope are tough to follow with diagonal mAb labels and epitope clusters. Given the result of 10-1074 in Fig 5E, how did that translate to binding differences (10-1074 is not in fig 6 or S14?).

We color-coded the legend in Fig. 6B (binding to mAbs) to show the epitope clusters, and used the same colors for the neutralization data that was added to the revised figure. While we did not test binding to 10-1074, we show neutralization by 10-1074 with increased sensitivity for the subtype C consensus with redesigned HV loops and for the CRF01_AE consensus with the shortest V1HV, but not for the CRF01_AE consensus with the V1HV that retained the conserved ‘IGNITD’ motif.

These results support that neutralization by 10-1074 is dependent on short V1HV rather than on the glycan at 332. This is described in the revised manuscript:

‘Finally, we evaluated how the Env consensus proteins (with or without modified HV loops) were neutralized by bnAbs. We used a pseudovirus neutralization assay in TZM-bl cells and tested ten bnAbs targeting the V2-apex (PG9 and PGT145), the glycan supersite (10-1074 and PGT128), the CD4 binding site (VRC01 and 3BNC117), the MPER (10E8 and 4E10) and the carbohydrate-

specific mannose-dependent (2G12) epitopes as well as the soluble CD4 (sCD4) (**Fig. 7EF**). There were differences in the neutralization sensitivity when the consensus Env with the redesigned HV loops were compared to those with the native HV loops, yet the differences were not consistently in the same direction (**Fig. 7EF**). Across the 10 bnAbs, there was no significant increase in neutralization sensitivity for subtype B and CRF01-AE with redesigned HV loops (one-tailed exact Wilcoxon-Pratt signed-rank test, $p > 0.201$). In contrast, the subtype C Env with redesigned HV loops was more sensitive to the ten bnAbs than the subtype C Env with unmodified loops ($p = 0.047$ for IC₅₀ values and $p = 0.016$ for IC₈₀ values). Moreover, CRF01_AE consensus proteins with the redesigned HV loop version 1 became sensitive to the glycan supersite bnAb 10-1074 (IC₅₀ = 0.544 µg/ml) and to the sCD4 (IC₅₀ = 0.617 µg/ml) while the original CRF01_AE consensus was resistant (IC_{50/80} > 50 µg/ml). Interestingly, the CRF01_AE consensus with redesigned HV loop version 1 was sensitive to 10-1074 while the Env with the version 2 of the redesign was resistant, in agreement with the antibody access predictions described above. The redesign version 1 is shorter in V1HV indicating that resistance to 10-1074 in CRF01_AE is associated with the V1 loop length and is not necessarily mediated by the lack of glycosylation at site 332 as neither redesigned consensus Env had a glycan at site 332.'

This is also discussed in the discussion:

'Interestingly, our structural modeling uncovered a novel mechanism for CRF01_AE's remarkable resistance to glycan-supersite antibodies such as 10-1074. It was previously proposed that 10-1074 resistance was mediated by a shift of the glycan at site 332 to site 334⁴⁹. Here, we showed that our redesigned Env CRF01_AE consensus with the shortest V1HV (without the 'IGNITD' motif) was sensitive to 10-1074 despite having a glycan shifted from site 332 to 334 (**Fig. 7EF**). The other (longer) redesigned CRF01_AE V1HV with the 'IGNITD' motif (also with no PNGS at site 332) was resistant to 10-1074, indicating that the long V1HV was associated 10-1074 resistance. In our model of CRF01_AE V1HV N-anchor sites, the I147 in the 'IGNITD' motif was predicted to occupy the same location as the CDRH3 of 10-1074 (**Fig. 6E** and **S3**), a pocket composed of sites 328, 330, 415 and 417. In contrast, the pocket was not occupied by the modeled consensus B or C Env with unmodified V1HV. Furthermore, PGT128, a V3-glycan antibody whose epitope does not occupy this pocket¹³ and is thus less occluded by the CRF01 AE V1HV, retains the capacity to neutralize half of the CRF01 AE viruses⁴⁹. Our neutralization results confirmed that all the Env consensus sequences were sensitive to PGT128, while only the CRF01_AE with the shortest V1HV was sensitive to 10-1074 (the two CRF01_AE Envs that retained the 'IGNITD' motif remained resistant to 10-1074) (**Fig. 7EF**).'

Reviewer #2 (Remarks to the Author):

Current HIV-1 vaccine designs using the prefusion closed form mainly focused on stabilization of the prefusion conformation, targeting specific germline precursors or glycosylation forms. The effects of variable loops which form or surrounding major sites of vulnerability are seldomly considered. The manuscript "Contemporary HIV-1 consensus Env with redesigned hypervariable loops promote antibody binding" by Bai and colleagues utilized from HIV-1 database, analyzed the length distribution of variable loops V1, V2, V4 and V5 in HIV-1 gp120 and the effects on plasma antibody and monoclonal antibody binding and neutralization, they employed AlphaFold modeling to model gp120 monomers with different length of variable loops. Based on these data, the authors designed HIV-1 Env with consensus sequences and altered hyper variable loops and showed that these designed proteins increased magnitude of binding responses to pooled plasma samples and representative bnAbs.

Overall, this is a well-designed study and the results added new information needed for the

design of an effective HIV-1 vaccine. The study could be improved if the authors could perform some of the additional characterization for the developed immunogens:

1. What are the conformation of the designed Env proteins? A simple negative EM with 2D and 3D classification might confirm the conformational states.

In the initial submission, we used uncleaved gp140s. As suggested by the reviewer, we ran negative EM with 2D and 3D classification for the gp140s presented in the initial manuscript as well as for stabilized trimers corresponding to our consensus Env sequences with or without modified loops. The gp140s are a mixture of trimers, dimers and monomers. These results are presented in the revised manuscript and we also show that all our computer-generated consensus sequences when produced as stabilized prefusion closed trimers had typical trimeric structures. The text was modified as follows:

'Next, we produced our consensus sequences as gp140 glycoproteins. Negative-stain electron-microscopy (EM) showed that gp140s for subtypes B, C and CRF01_AE were mostly open trimers and monomers, with only monomers being averageable from negative stain EM (**Fig. S14**). Finally, we stabilized Env in the prefusion conformation by combining previously reported modifications^{47,48} and three extra mutations. As stabilized trimers, produced by transient transfection and purified by GNA-lectin and size-exclusion chromatography, consensus B and C Env were just under 50% trimer by 2D averaging of negative-stain EM data (42 and 45%, respectively); the HV loop redesign marginally increased the population of trimer observed to 59 and 55%, respectively (**Fig. 5B**). To determine whether the HV loop redesign affected the opening of the trimer, we used 3D classification and refinement without any applied symmetry. For the subtype B consensus, 76% of trimers classified with the reconstruction identifiable as the closed prefusion state, while 24% appeared open, but still in a prefusion state (**Fig. 5C**). Results were similar for the HV loop redesign with 68% in the closed and 32% in the open prefusion state. In contrast, the subtype C consensus trimers were only at 31% in the closed prefusion state with 69% appearing open, with minimal changes with the HV loop redesign (29% in the closed and 71% in the open prefusion state). We cannot rule out that the conformational landscape of stabilized trimers is affected by the production and purification process as well as the adsorption to the carbon grid, which may explain the substantial fraction of monomers observed by negative-stain EM. Overall, these result suggest that the HV loop redesign marginally increased the trimer yield and had limited impact on the conformation of Envs.'

2. How is the glycan shield looking after modifying the HVs? Could the authors perform a glycan analysis on the expressed trimer?

We agree that this would be interesting information but we do not have these data for this revised manuscript. We have prefusion closed trimers and we will plan to conduct glycan analysis in future studies to characterize the effect of the loop modifications.

3. Are these Env expressed without commonly used SOSIP stabilization?

Here we used gp140s to initially verify the designs, no SOSIP stabilization mutations were included. We have now generated sequences that include repair and stabilization mutations to create stabilized trimers. These results are reported in the revised manuscript.

Details on the sequence design are presented in the methods:

'Stabilized, soluble trimers in a prefusion state were engineered by introducing previously reported^{47,48} and additional modifications with ten stabilization mutations (M302, V304, L320, N535, P556, E588, V589, F651, I655 and V658) and three SOSIP mutations (C501, C605 and P559). Sites 508-511 were replaced with six arginines for more efficient furin cleavage. All Envs were terminated at site 664.'

Figures were added in the revised manuscript to show the EM data for all tested proteins (**Fig. 5 and Fig. S14**).

4. The authors used R to S mutations to disable furin cleavage. How did the authors make sure that the C-term of gp120 and N-term of gp41 are in correct locations? We know from the prefusion closed form that the end of gp120 and the N-term of fusion peptide are located apart. As mentioned above, we created trimers and present the EM data that support the proper conformation of our designed Env.

Reviewer #3 (Remarks to the Author):

Bai et al. report a new modeling-based strategy for designing Envs with improved bnAb binding profiles, focusing on shortening the hypervariable loops V1, V2, V4, and V5. The authors rely on AlphaFold2 for building models of gp120s for designed Envs with shortened hypervariable loops from 3 different clades of HIV and test a subset of the designs with experimental binding assays. The results presented could be informative for guiding the design of hypervariable loops in immunogens, however, it is hard to come to definitive conclusions about the data as it is currently presented. My comments focus on clarifying statistical results:

1. In the boxplots in Figure 1, the meaning of the whiskers is not defined. This needs to be clarified in the figure legends so that the figures can be accurately interpreted by the reader. We thank the reviewer for this suggestion. We revised the figures and the legends of all figures with a distribution to clarify that the median, minimum, maximum and 25th/75th percentiles are depicted with box plots.
2. Figure 2 and related supplementary figure 1 does not define the meaning of the whiskers in the boxplots. Furthermore, supplementary figure 1 does not describe the boxes, centrality, or distributions shown in the boxplots. This is now defined in all figures.
3. For supplementary figures related to figures 3 and 4, some are missing either the N-anchor or C-anchor and it is unclear why those are not needed for some of the designs. Explaining why N/C-anchor was/were not used would help to clarify the design strategy. We considered the N/C-anchor as the relatively conserved N/C-terminal sites (relative to other HV sites) that might form interactions with the non-HV part. When the N/C-anchor was missing in the HV-loop redesign, the non-HV part of Env at the N/C-terminals of the loop served a similar role. We provided specific descriptions in the figure legends in the revised supplementary material.
4. It's unclear where the replicate values in Figure 5B-D are coming from. We thank the reviewer for this comment and realized that this analysis may have caused confusion and describe a new analysis in the revised paper. The different points in Figure 5B-D (now Fig. 6) were not from replicate values but corresponded to different sites in each epitope. For example, there are 15 sites in the epitope of 10-1074 and 35 in the epitope of VRC01, so there were 15 and 35 points in the plot for 10-1074 and VRC01, respectively. This analysis is removed. We conducted a different analysis which better integrates the flexibility of the loop conformation: we analyzed the epitope depth for different modeled loop conformations using the KIC loop modeling protocol of Rosetta and present these data in the revised Fig. 6 (previously Fig. 5). We consider that this method is more robust but the conclusion is identical: 'The epitope depth decreased for the four bnAb epitopes of the Env consensus of subtypes B, C, and CRF01 AE (Fig. 5B-D). The median depth decreased by 0.8 to 1.6 Å for glycan supersite

antibody epitope, while the median depth decreased by 0.2 to 0.8 Å for CD4bs antibody epitope.'

5. Figure 2D V1HV long is missing the whiskers and there is no explanation. Whiskers are also missing from supplementary figure 1A PGT145-other V2HC long, 1B VRC26.25-B V2HV short and long and V5HV long, and 1D PGT135-C V1HV short and long. Add an explanation of the whiskers and why they are missing from some plots.

We thank the reviewer for the careful review as we had failed to define the whiskers. The upper whisker was drawn from the upper quartile (Q3) to the largest observed data point that was below $[Q3+1.5IQR]$ and there was no data point in that range for the case mentioned above, hence there was no whisker. There was a parallel situation for the lower whisker. We realized that this was confusing and revised the figures with a more standard boxplot visualization that includes median, minimum, maximum and 25th and 75th percentiles of the distribution.

6. In figures 2B, 2C, 4A, and supplementary figures 1A, 1B, 1C, 1D, 1E, and 1F some of the whiskers do not include the full spread of the data, which can only be if they considered those points outliers but outlier detection methods were not described in the methods or figure legends.

This was revised throughout the manuscript as mentioned above.

7. In Figure 2 and supplementary Figures 1 and 2 all IC50 values over 100 ug/mL are grouped. However, this makes interpreting the boxplots difficult. The position of the median does not make sense in some cases because $\geq 50\%$ of the points occur at lower IC50 values than where the median is marked. This trend occurs more often in the long-loop box plots than in the short-loop plots. Plot the full distribution, including on $IC_{50} \geq 100 \text{ug/mL}$, so that the reader can more easily interpret the statistics described by the boxplots.

We agree with the reviewer that it would be better to plot the full distribution. However, the IC50 data used for these figures are obtained from the CATNAP database and the IC50 measurements in CATNAP are capped at 100 ug/ml. We clarified this in the figure legends.

8. There are no methods for outlier detection/removal listed but it seems like outliers were excluded. In some of the boxplots, the whiskers do not include the full range of data, suggesting that these points were excluded as outliers. This is only true for points with low IC50 values and this occurs more frequently in Envs with long VH loops. Potential outliers with high IC50 values do not seem to have been considered because all values over 100ug/mL are grouped. If only low IC50 values were analyzed for outliers, without checking for high outliers, making the outlier detection one-sided, this skews the entire analysis towards higher IC50 values. Since this occurs more frequently in Envs with long VH loops I am concerned that the analysis skews the results towards supporting the hypothesis that shorter loops improve bnAb binding. Providing a description of the outlier detection and removal process and plotting the full distribution of the points $\geq 100 \text{ug/mL}$ would strengthen the arguments and provide more robust support for the claims that shorter HV loops improve bnAb binding.

We did not remove any outliers. We realize there is a limitation with the CATNAP upper limit of IC50 values at 100 ug/ml, and addressed this issue as described above.

9. Lines 175-176 state that D133 interacts with K154 but is interacting with K155 in Figure 3. We thank the reviewer for identifying this error. It should be K155, and K154 was a typo. We corrected the text.

10. The section describing the HV loop redesign is vague and lacks critical information. Some of the interactions discussed do not make interactions in the structures presented in the figures.

Was more than one conformation used to calculate the structures? Other potential interactions shown in the figures are not discussed in the text (ex: K151 L134/D133). Discussion of important interactions considered in the redesign of other loops is also missing. Furthermore, it is unclear how the length of the linker in the design was chosen. The linker is different for every designed construct but there is no information about how the specific number of amino acids was chosen. The missing details obfuscate how the design strategy was carried out. Adding a more thorough discussion of the interactions and reasoning behind the linker design used in the HV design strategy would greatly strengthen the claims and importance of the findings.

The design was done manually with the assistance from AlphaFold2. The general rule is to keep the loop short while maintaining the integrity of the structure but the exact loop length is determined on a case by case basis by testing different modifications.

When loops of different lengths were tested, we selected the redesigned loop that showed a similar or higher predicted IDDT score considering the anchor and surrounding sites (to ensure there is an undisturbed surrounding area when the loop is replaced). We clarified this in the text: 'To shorten the spacer, we evaluated different lengths iteratively using AlphaFold2 predicted structures by comparing the predicted local Difference Distance Test (IDDT) scores for the anchor and surrounding sites of V1HV. A specific spacer is selected when it maintains or increases the predicted IDDT (i.e., preserving the anchor stability).'

We also added additional details in the legend of the different figures.

11. AlphaFold2 still struggles to provide accurate structures for unstructured loop regions in proteins, especially for domains that can exhibit multiple conformations, yet this method was used to model the flexible and unstructured HV loops in Env. The structures used in this accessibility analysis may not be reliable. The fit scores from AlphaFold2 should be provided so that the reader can assess the quality/validity of the models.

We thank the reviewer for this suggestion. We added a new supplemental figure that shows the fit scores (predicted IDDT score) of the AlphaFold2 model for both the original and redesigned Env. We added the following sentence in the Results:

'The redesigned HV loops tended to have higher predicted IDDT scores from AlphaFold2 than the unmodified ones, except for the V4HV of the subtype C consensus (Fig. S13).'

We revised the accessibility analysis by remodeling the loops 500 times using the KIC loop modeling protocol of Rosetta. The revised Fig. 5 shows the mean depth for the different epitopes in the 100 models with the lowest energy (as evaluated by Rosetta Energy scores).

The text was modified as follows:

'We remodeled each loop 500 times via the KIC loop modeling protocol⁴⁷ from PyRosetta⁴⁸. Among the 500 models, the 100 models with the lowest Rosetta energy were selected for the epitope depth calculation.'

12. A sphere probe was used to assess the antibody accessibility in redesigned HV loop models, and this was followed up with binding studies on a select few designs, however, there was no correlation shown between the modeling and experimental results. One analysis that could strengthen this paper would be to include a regression analysis between the antibody accessibility predicted by the modeling and the experimental antibody binding data. Showing a correlation between these two data sets would establish the predictive value of the modeling and design strategy.

We thank the reviewer for this suggestion.

We analyzed this relationship for three antibodies 3BNC117, VRC01 and PGT128, as both Ab:Env complex structures and antibody binding data are available. While there are also Ab:Env complex structures for PG9 and PGDM1400, these Abs target the apex of the prefusion Env

trimer and since we modeled only one subunit of the trimer, these Abs are not included in the analysis. We found no significant correlation between the predicted epitope depth change and the antibody binding signal change (Spearman's $\rho=0.52$, $p\text{-value}=0.15$). There could be many reasons for the lack of association, including that the epitope depth may not be quantitatively correct in capturing the antibody accessibility. This comparison was added in the Results along with the Supplementary figure pasted below.

Fig. S17. Comparison between the improvement of antibody accessibility and the improvement in antibody binding. The epitope depth corresponds to the mean value of 100 lowest energy models using loops modeled with the KIC loop modeling protocol of Rosetta.

REVIEWER COMMENTS

Reviewer #1 (Remarks to the Author):

Bai et al 2nd review

In this reworked version, Bai et al use alphafold to modify consensus Env sequences at their variable loops with the stated plan to try to eliminate hypervariable parts of the trimer that may shield conserved bNAb epitopes and eliminate the unwanted type-specific responses that they may elicit instead of bNAbs.

This version is a great improvement. This has gone from being a good paper to an important one. Some comments are:

Fig 2 is really interesting. It's a pity not to use some colors in this fig? to bring out the patterns that were significant, perhaps? $P < 0.05$. Possibly subtypes could color the dots so subtype differences can be seen (though S1 Figure covers this also?). There is quite a bit to process here for the reader. Colors could also be used to distinguish the short loop and long loop segments of the distributions in the first column? Of Fig. S2?

Comment: It occurs that loop lengths might be linked in strains. So for example, a long V1 HV might be linked to long V2 HV, allowing the top of the spike to fold. Is there a way in this analysis of checking for such possible linkages? This would help determine what might be "allowable" in terms of making deletions/insertions, i.e., should changes in V1HV be made at the same time as those in V2HV to better preserve folding? Also, somewhat related to this, it is notable that some strains exhibit no HV sequence. Do these short loop strains just exhibit one short loop or do they occur together? Based on Table 1, it appears this may not be the case. Lengths are essentially random? Are these strains any more sensitive than garden variety strains. It raises the question of whether there is some value in poring over all these natural sequences and checking some of interest – without modifying them as a parallel strategy for immunogen selection to that described.

If the figures are ordered BCE, then it seems Fig S3 and S4 are out of order – should be CE not EC.

V4HV C to W seems a big change. Is the C that is replaced involved in a disulfide in this strain?

It probably should be explained in the text what the meaning of high and low IDDT tests means..

Re: pseudoviruses, we can truly gauge the effects of the changes in a functional format. In particular, it would be useful to know if the trimers maintain a "closed" format as in the unmodified version by checking sensitivity to V3 non-nAbs (that cross-react with the V3 loop in a gp120 format). Alternatively non-neut sera could be used if it contains V3 antibodies that cross-react with V3 peptides of the strain in question. I think more could be made of the heat maps in Fig 7E, if plotted as dot plots or histograms or perhaps even better, the delta changes for each mAb. It looks like PG9 fares well here, I think this is almost the center of the paper to see where the strategy is gaining ground or where it is static. PGT145 is not affected, but this is a mAb that targets the top of the apex rather than the side of it (Rantalainen/Ward), so the PGT145 epitope may not be subject to as much clashes as PG9, considering its approach angle? It is difficult to process the nice data in Fig 7E without having the benefit of a visual (aside from the colors which help a bit). This neut data may be interpreted as telling us how much of a problem these HV loops are for these prototype bNAbs. These bNAbs have evolved already to a point where they can tolerate a lot of clashes, by focusing on conserved contacts. This being the case, it is not unreasonable that shortening and remodeling HV might not make them bind better as they have evolved to overcome these binding constraints. So if the IC50s are not improved but in the same ballpark, that would be a good result too. The redesigns are therefore not damaging epitopes and are hopefully providing a more sensitive version of these canonical epitopes that might be useful in

inducing them, by eliminating clashes that might occur with germline-reverted bnAbs, if not with bnAbs themselves. The CD4 KI with AE is also interesting. I really don't think the heat map alone does this data justice. If clade C viruses are more sensitive to all with the redesign, that merits a graph! How does the neut data jive with the binding data. Similar patterns or not?

The text on depth on page 8 seems to be covered again on page 9 (Fig 6 e f).

I am not sure the complete sequences are recorded? HV regions are recorded, but not the new consensus sequences. Are there notable changes compared to previous consensus sequences that have been used based on earlier sequences?

The neut results could be amplified in the abstract. Also the AE neut by 10-1074 being sensitive to loop length not lack of N332 could be mentioned in the abstract.

Reviewer #2 (Remarks to the Author):

The revised manuscript is much improved. The added negative stain EM on the designed HIV-1 Env and classification of Env in different states are great result to include. I have no further comment.

Reviewer #3 (Remarks to the Author):

Bai et al. addressed all comments and added material where requested satisfactorily.

Please find our responses in blue below.

Reviewer #1 (Remarks to the Author):

Bai et al 2nd review

In this reworked version, Bai et al use alphafold to modify consensus Env sequences at their variable loops with the stated plan to try to eliminate hypervariable parts of the trimer that may shield conserved bNAb epitopes and eliminate the unwanted type-specific responses that they may elicit instead of bNAbs.

This version is a great improvement. This has gone from being a good paper to an important one. Some comments are:

Fig 2 is really interesting. It's a pity not to use some colors in this fig? to bring out the patterns that were significant, perhaps? $P < 0.05$. Possibly subtypes could color the dots so subtype differences can be seen (though S1 Figure covers this also?). There is quite a bit to process here for the reader. Colors could also be used to distinguish the short loop and long loop segments of the distributions in the first column? Of Fig. S2?

We thank the reviewer for these suggestions. Figures 2, S2 and S3 were regenerated accordingly, as shown below.

Comment: It occurs that loop lengths might be linked in strains. So for example, a long V1 HV might be linked to long V2 HV, allowing the top of the spike to fold. Is there a way in this analysis of checking for such possible linkages? This would help determine what might be “allowable” in terms of making deletions/insertions, i.e., should changes in V1HV be made at the same time as those in V2HV to better preserve folding? Also, somewhat related to this, it is notable that some strains exhibit no HV sequence. Do these short loop strains just exhibit one short loop or do they occur together? Based on Table 1, it appears this may not be the case. Lengths are essentially random? Are these strains any more sensitive than garden variety strains. It raises the question of whether there is some value in poring over all these natural sequences and checking some of interest – without modifying them as a parallel strategy for immunogen selection to that described.

We evaluated whether the hypervariable loop lengths of V1 and V2 were linked. We found no association between V1HV and V2HV lengths, as shown by very low values for Spearman’s Rho coefficients in the figure below. The most frequent V1HV and V2HV length combinations shown in yellow reflect average lengths. Only a minor population of sequences had very short V1 and V2 hypervariable loops. We added this figure to the supplement as Fig S1.

Fig. S1. Lack of correlation between HIV-1 Env hypervariable loop lengths for V1HV and V2HV. The distribution of loop lengths is shown for subtypes B and C, and CRF01_AE with Spearman’s correlation coefficient labeled. The number of sequences of identical length are color-coded from dark blue (minimum number, n=1) to yellow (maximum number, n=92, 23, 29 for subtype B, C and CRF01_AE, respectively) after a log transformation.

In addition, we checked whether the sequences with the shortest combination of V1HV and V2HV were driving patterns seen with the lengths of V1HV or V2HV. In the two figures below that show the sensitivity to PGT145 and VRC26.25, sequences with the shortest V1HV+V2HV (shortest 5% of the distribution) are colored in orange and sequences with the longest V1HV+V2HV (top 5% of the distribution) are colored in blue. There was no clear evidence that the sequences with the shortest V1HV+V2HV dominated among sensitive strains.

If the figures are ordered BCE, then it seems Fig S3 and S4 are out of order – should be CE not EC. We thank the reviewer, this was corrected. The Figures S3 and S4 were reordered (now Fig. S4 and S5).

V4HV C to W seems a big change. Is the C that is replaced involved in a disulfide in this strain? That is correct, the C is predicted to form a disulfide bond in around 25% of CRF01_AE sequences, and we replaced it with W. We added one sentence in the manuscript. ‘One notable feature of the V4HV redesign for CRF01_AE is that C395 (which was predicted to form a disulfide bond in our sequence) was replaced with the second most frequent residue at that site, W, and the spacer was extended to better shield the surrounding sites of V4HV (Fig. S13).’

It probably should be explained in the text what the meaning of high and low IDDT tests means..

We revised the text on page 6 as follows:

‘To shorten the spacer, we evaluated different lengths iteratively using AlphaFold2 predicted structures by comparing the predicted local Difference Distance Test (lDDT) scores, which measure the confidence given by AlphaFold2 for each output structure. A specific spacer was selected when it maintained or increased the predicted lDDT for the anchor and surrounding sites of the HV loop, suggesting that the anchor stability will be maintained.’

Re: pseudoviruses, we can truly gauge the effects of the changes in a functional format. In particular, it would be useful to know if the trimers maintain a "closed" format as in the unmodified version by checking sensitivity to V3 non-nAbs (that cross-react with the V3 loop in a gp120 format). Alternatively non-neut sera could be used if it contains V3 antibodies that cross-react with V3 peptides of the strain in question.

We consider that the neutralization data with PGT145 can partially address this question. Since PGT145 recognizes the closed prefusion trimer, we confirmed that the redesigned HV loops can still form closed prefusion trimers, as also shown with the cryo-EM data. We agree that binding to V3-loop or V3-peptide specific antibodies (e.g. 447-52D) can provide more information about the conformation of Env, including regarding what is the relative abundance of Env that are not in a closed prefusion conformation, as the V3-peptide is not easily accessible to antibodies in the closed-prefusion Env. We initiated such experiments but further studies will be needed to better investigate the results.

We added the text below in the Results section and added a Supplementary figure (S19):

‘In addition, we transfected 293T cells with Env-gp160 plasmids corresponding to unmodified and redesigned HV loops and evaluated bnAb binding. For the two subtype B consensus sequences, we showed comparable Env cell surface expression and staining with bnAbs such as 3BNC117 and PGDM1400 indicated that Env were expressed as closed trimers. Moreover, seven bnAbs tested showed increased binding to the Env with redesigned HV loops, suggesting improved epitope display (Fig. S19).’

Fig. S19. BnAb epitope recognition for HIV-1 subtype Env consensus sequences with or without HV loop modifications. Eight bnAbs were used to compare the cell surface expression of Env consensus sequences with unmodified (red) and redesigned (blue) HV loops. 293T cells transfected with DNA expression constructs were stained with bnAbs and analyzed by flow cytometry. Median fluorescent intensity values are indicated; 2G12 reference staining confirms comparable protein expression.

I think more could be made of the heat maps in Fig 7E, if plotted as dot plots or histograms or perhaps even better, the delta changes for each mAb. It looks like PG9 fares well here, I think this is almost the center of the paper to see where the strategy is gaining ground or where it is static. PGT145 is not affected, but this is a mAb that targets the top of the apex rather than the side of it (Rantalainen/Ward), so the PGT145 epitope may not be subject to as much clashes as PG9, considering its approach angle? It is difficult to process the nice data in Fig 7E without having the benefit of a visual (aside from the colors which help a bit). This neut data may be interpreted as telling us how much of a problem these HV loops are for these prototype bnAbs. These bnAbs have evolved already to a point where they can tolerate a lot of clashes, by focusing on conserved contacts. This being the case, it is not unreasonable that shortening and remodeling HV might not make them bind better as they have evolved to overcome these binding constraints. So if the IC50s are not improved but in the same ballpark, that would be a good result too. The redesigns are therefore not damaging epitopes and are hopefully providing a more sensitive version of these canonical epitopes that might be useful in inducing them, by eliminating clashes that might occur with germline-reverted bnAbs, if not with bnAbs themselves. The CD4 KI with AE is also interesting. I really don't think the heat map alone does this data justice. If clade C viruses are more sensitive to all with the redesign, that merits a graph! How does the neut data jive with the binding data. Similar patterns or not?

We followed these suggestions to modify figure 7, which now includes only the IC50 data for the neutralization results and a new Supplementary figure was added to show the IC80 data (**Fig. S20**). Panel 7F shows a dot plot of the IC50 values separated by subtype, while panel 7G shows the data separated by bnAb.

Fig. 7. Fig. 7. Improved antibody binding to subtype B and CRF01_AE consensus Env with redesigned HV loops. Consensus Env sequences were expressed as gp140 glycoproteins. Comparison of the binding of (A) plasma pools from 13 cohorts of PLWH and (B) monoclonal antibodies to consensus B Env with unmodified (in red) and redesigned (in blue) HV loops. The number of plasma samples in each pool is shown in parenthesis in panel A. Binding of plasma samples (C) and monoclonal antibodies (D) to the subtype B, C and CRF01_AE consensus Env with unmodified HV loops (red) and redesigned HV loops (blue) is compared using paired, non-parametric one-tailed Wilcoxon signed-rank tests. MPER antibodies and PGDM1400 were not included in the statistical tests as their epitopes are not presented on the gp140 glycoproteins. (E) Neutralization sensitivity of the consensus Env with or without redesigned HV loops for subtypes B, C and CRF01_AE measured for ten bnAbs that targeted the CD4 binding site (VRC01 and 3BNC117), V2-apex (PG9 and PGT145), glycan supersite (10-1074 and PGT128), MPER (10E8 and 4E10) and mannose-dependent (2G12) epitopes as well as the sCD4. The neutralization sensitivity is reported as IC50 values ($\mu\text{g/ml}$) with the most sensitive viruses colored in red and the most resistant in gray. Neutralization sensitivity (IC50 values) shown for the unmodified and redesigned HV loops and displayed separately for each Env clade (F) or each bnAb (G). The increase in sensitivity with the redesigned HV loops is tested for each clade with one-tailed exact Wilcoxon-Pratt signed-rank tests. Only the 01_AE_2010s.hv_redesigned.1 is shown in panels F and G as it has a shorter V1HV than 01_AE_2010s.hv_redesigned.2.

We modified the text in the Results section to mention the data for PG9 as the change in sensitivity is clear for the three redesigned Env:

‘One bnAb, PG9, showed a notable increase in sensitivity for the antigens with redesigned HV loops with a similar pattern across subtypes/CRF (Fig. 7G).’

We also added some language in the discussion.

‘There were nonetheless striking effects regarding the increased sensitivity to PG9 across clades or the change from resistance to sensitivity to 10-1074 for CRF01_AE Env. These results illustrate that Env sensitivity to neutralization is not an all or none phenomenon and warrant further studies.

Although a significant increase in sensitivity was not evidenced across all bnAbs and all clades, it remains possible that our redesigned Env when used as immunogens could promote the immunofocusing of bnAb responses to conserved epitopes due to the elimination of potentially interfering residues.’

We tested the association between the binding and IC50 data and only found weak correlations. While the trend is generally in the right direction (negative correlation), we think our comparison could be affected by two factors:

- Only 7 bnAbs were shared across binding and neutralizing datasets (VRC01, 3BNC117, PG9, PGT128, 2G12, 4E10, 10E8)
- The binding data is based on gp140 monomers, while there were Env trimers for the neutralization

While there is not a significant increase in sensitivity across all bnAbs tested, we hope we have also conveyed that the redesigns did not negatively impact the neutralization sensitivity. We plan to conduct more studies to better understand these patterns. For example, our modifications are distant from the MPER region so we did not expect to see changes in sensitivity for MPER antibodies, yet there are some differences.

The text on depth on page 8 seems to be covered again on page 9 (Fig 6 e f).

We reorganized that part of the Results.

I am not sure the complete sequences are recorded? HV regions are recorded, but not the new

consensus sequences. Are there notable changes compared to previous consensus sequences that have been used based on earlier sequences?

The Env sequences used for the HV loop redesign correspond to the newly designed Env we derived by analyzing publicly available sequences for the main subtypes/CRF. The manuscript describing this work is titled 'HIV-1 Gag, Pol, and Env diversified with limited adaptation since the 1980s' and it was recently published in mBio: <https://doi.org/10.1128/mbio.01749-23>

The unmodified consensus sequences can be found in the Supplement of the manuscript or on our webpage. We added the reference in the revised manuscript.

The new Env consensus sequences show some changes but it is not a drastic change of the consensus. My understanding is that the consensus sequences used for experimental assays usually correspond to the consensus sequences that were made available in 2004 by LANL. LANL published a new set of consensus sequences in 2021, based on all sequences sampled until 2019. Our approach to designing consensus sequences differed from LANL's approach as we decided to design a consensus sequence specific to each decade. For our study, we used the consensus sequences derived from all the sequences sampled after 2010 to prioritize recently circulating viruses.

The neut results could be amplified in the abstract. Also the AE neut by 10-1074 being sensitive to loop length not lack of N332 could be mentioned in the abstract.

We thank you for this suggestion and modified the abstract as follows:

'Neutralization assays showed increases in sensitivity to bnAbs, although not always consistently across clades. Strikingly, the HV loop modification rendered the resistant CRF01_AE Env sensitive to 10-1074 despite the absence of a glycan at N332.'

We are also wondering about modifying the title to 'Contemporary HIV-1 consensus Env with AI-assisted redesigned hypervariable loops promote antibody binding'

We are very grateful for all the useful and thoughtful suggestions.